# Robust, reproducible and quantitative analysis of thousands of proteomes by micro-flow LC–MS/MS

Yangyang Bian[1,2], Runsheng Zheng[1], Florian P. Bayer[1], Cassandra Wong[3], Yun-Chien Chang[1], Chen Meng[4], Daniel P. Zolg[1], Maria Reinecke[1,5,6], Jana Zecha [1], Svenja Wiechmann [1,5,6], Stephanie Heinzlmeir[1], Johannes Scherr[7], Bernhard Hemmer [8,9], Mike Baynham[10], Anne-Claude Gingras [3], Oleksandr Boychenko[11] & Bernhard Kuster [1,4,5]*

Nano-flow liquid chromatography tandem mass spectrometry (nano-flow LC–MS/MS) is the mainstay in proteome research because of its excellent sensitivity but often comes at the expense of robustness. Here we show that micro-flow LC–MS/MS using a 1 × 150 mm column shows excellent reproducibility of chromatographic retention time (<0.3% coefficient of variation, CV) and protein quantification (<7.5% CV) using data from >2000 samples of human cell lines, tissues and body fluids. Deep proteome analysis identifies >9000 proteins and >120,000 peptides in 16 h and sample multiplexing using tandem mass tags increases throughput to 11 proteomes in 16 h. The system identifies >30,000 phosphopeptides in 12 h and protein-protein or protein-drug interaction experiments can be analyzed in 20 min per sample. We show that the same column can be used to analyze >7500 samples without apparent loss of performance. This study demonstrates that micro-flow LC–MS/MS is suitable for a broad range of proteomic applications.

---

[1] Chair of Proteomics and Bioanalytics, Technical University of Munich, Freising, Germany. [2] Medical Research Center, The First Affiliated Hospital of Zhengzhou University, Zhengzhou, China. [3] Lunenfeld-Tanenbaum Research Institute, Sinai Health System, Toronto, ON, Canada. [4] Bavarian Biomolecular Mass Spectrometry Center (BayBioMS), Technical University of Munich, Freising, Germany. [5] German Cancer Consortium (DKTK), partner site Munich, Munich, Germany. [6] German Cancer Research Center (DKFZ), Heidelberg, Germany. [7] Centre for Preventive and Sports Medicine, Klinikum Rechts der Isar, Technical University of Munich, Munich, Germany. [8] Department of Neurology, Klinikum Rechts der Isar, Medical Faculty, Technical University of Munich, Munich, Germany. [9] Munich Cluster for Systems Neurology (SyNergy), Munich, Germany. [10] Thermo Fisher Scientific, Runcorn, UK. [11] Thermo Fisher Scientific, Germering, Germany. *email: kuster@tum.de

Nano-flow liquid chromatography (nano-flow LC) has been the mainstay in proteome research for >20 years[1], primarily because low flow rates improve peptide ionization by electrospray (ESI) for mass spectrometry (MS) and, hence, sensitivity[2,3]. However, this comes at the cost of the challenge of manufacturing reproducible and long-lasting columns, maintaining stable ESI over extended periods of time, rapid chromatographic overloading, mass spectrometric saturation and often long, unproductive overhead times for sample transfer at low flow rates[4–6]. These factors can limit the reproducibility of peptide identification and quantification as well as the comprehensiveness, robustness and throughput of proteome analysis, particularly when analyzing samples of high complexity or wide dynamic range of protein concentrations as represented by tissues and body fluids[7].

Particularly for targeted quantitative MS assays such as selected reaction monitoring (SRM)[8] or, more recently, parallel reaction monitoring (PRM)[9], where the mass spectrometer is focused on a small number of analytes to maximize sensitivity and quantitative accuracy/precision[10], standard analytical HPLC columns (2.1 mm inner diameter, ID) are frequently used[11–15] to address the aforementioned challenges. Also for so-called data-independent acquisition (DIA)[16] methods that aim to catalog the peptides present in a sample systematically, the field is increasingly adopting 300 μm ID columns as a compromise between sensitivity and robustness[17,18].

As the sensitivity of mass spectrometers has greatly improved over the years as a result of e. g. more efficient ionization[19], ion transfer and detection[20–23], further advances in untargeted (also referred to as discovery-type) and data dependent (DDA) proteome analysis may be sought by improving peptide separations[24–26]. For example, Gonzalez et al.[27] employed a standard 2.1 mm ID analytical column to identify about 800 proteins and 4,000 peptides from 40 μg E. coli protein digest using a 120 min LC gradient. However, sample quantities of that order may often not be available from biological sources. More recently, Lenčo et al.[28] reported the identification of about 2,800 human proteins in 60 min from 2 μg HeLa protein digests using an online LC–MS/MS method employing a 1 mm ID column. In a series of elegant experiments, that report demonstrated that discovery proteomics is feasible in principle using such a micro-flow LC–MS/MS system. Another recent interesting approach was presented by Bache et al.[29] who introduced specialized new chromatographic hardware that aims to combine the advantages of micro-flow and nano-flow LC. Here, complex digests were separated at flow rates of 10–20 μl/min at very low pressure using stage tips[30], embedded in a pre-formed LC-gradient and subsequently analyzed by online nano-flow LC–MS/MS. The authors showed that the system identified nearly 10,000 human proteins and 130,000 peptides from fractionated HeLa protein digests within 18 h and that the system is stable across over 2,000 injections.

Here, we report on the systematic evaluation of the merits of online micro-flow LC–MS/MS for quantitative discovery proteome analysis using standard HPLC equipment available in any analytical laboratory. At the heart of the method is a commercial 1 × 150 mm reversed phase HPLC column operating at a flow rate of 50 μl/min coupled online to a sensitive and rapid mass spectrometer. Data collected from >2,000 samples show that most of the limitations of nano-flow LC can be overcome at a very moderate loss of practical sensitivity. The approach markedly improves robustness, throughput and reproducibility of quantification without the need for specialized equipment. The results suggest that this approach has the potential to transform the field because of the ease of its technical implementation, the wide range of feasible applications and the very high data quality which makes the system suitable for the analysis of clinical specimen.

## Results and discussion

**Basic performance characteristics of micro-flow LC–MS/MS.** The cross-sectional area of a 1 mm ID mico-flow LC column is 178 times larger than that of a 75 μm ID nano-flow LC column typically used in proteome research and the optimal flow-rate scales in the same way (Fig. 1a). While a wider column diameter improves separation efficiency by eliminating column overloading, the higher flow rate needed for a 1 mm ID column compared to a nano-flow LC column massively dilutes analyte concentration which should lead to a strong loss of electrospray ionization (ESI) efficiency and, as a result, sensitivity. We found that this can be partially off-set by the very narrow LC peaks afforded by the higher flow rate which increases peptide concentration (Fig. 1b, c and Source Data File) and by adding traces of DMSO that we have shown to enhance peptide ionization[19] (Supplementary Fig. 1 and Source Data File). As a result, only ~5× more sample was required on the micro-flow compared to the nano-flow LC system when using a 28 Hz MS data acquisition method to obtain similar numbers and quality of peptide and protein identifications in single-shot analysis of complex HeLa protein digests while maintaining superior chromatographic performance throughout (Fig. 1d–g and Source Data File). The faster 41 Hz MS data acquisition method available on the Orbitrap HF-X may also be used but requires ~10× more material in the micro-flow setup vs nano-flow LC–MS/MS (Supplementary Fig. 2 and Source Data File), which is why all of the data presented below (except for the TMT analysis, see methods) was collected using the 28 Hz method. A serial dilution analysis of the same HeLa protein digest showed that >1000 proteins could be identified from 200 ng of protein digest (quantified on protein level) when optimizing LC gradient times (30 min) and MS data acquisition parameters (28 Hz, Supplementary Fig. 3 and Source Data File) which is sufficient for a wide range of proteomic applications (see also below). We next tested the micro-flow system for the deep characterization of proteomes by off-line fractionation of digests using high pH reversed phase HPLC. From 200–400 μg of a HeLa or human placenta protein digest respectively (quantified on protein level), we identified nearly 10,000 proteins and between 120,000 and 140,000 peptides within 16 h of total analysis time. This is very comparable in quantity and quality to results from the recent nano-flow LC literature[29,31] (Fig. 2a, b and Source Data File). We note that digests in these reports were quantified on the peptide level. Hence, the actual differences between the quantities used between the different laboratories are likely smaller than the stated values. Because the micro-flow setup allowed for direct sample injection onto the column at 100 μl/min, the overhead times needed for sample loading, column equilibration etc. could be drastically reduced compared to popular nano-flow LC systems operating at low flow rates. This, in turn, increases the effective use of the mass spectrometer particularly for short gradient times, which supports the analysis of up to 96 samples per day (Fig. 2c, d).

**Deep-scale analysis of TMT multiplexed proteomes.** Next, we set up two independent micro-flow LC systems on an Orbitrap Q Exactive HF-X and an Orbitrap Fusion Lumos and found very similar performance and overlap for both label-free single shot and deep-scale proteomics analysis (Supplementary Fig. 4). A multiplexed and deep-scale analysis of 11 human cancer cell lines using tandem mass tags (TMT) resulted in identification of ~7800 and ~6400 proteins using MS2 (HF-X) and MS3 (Lumos) methods, respectively, from 250 μg of peptides within 16 h of total

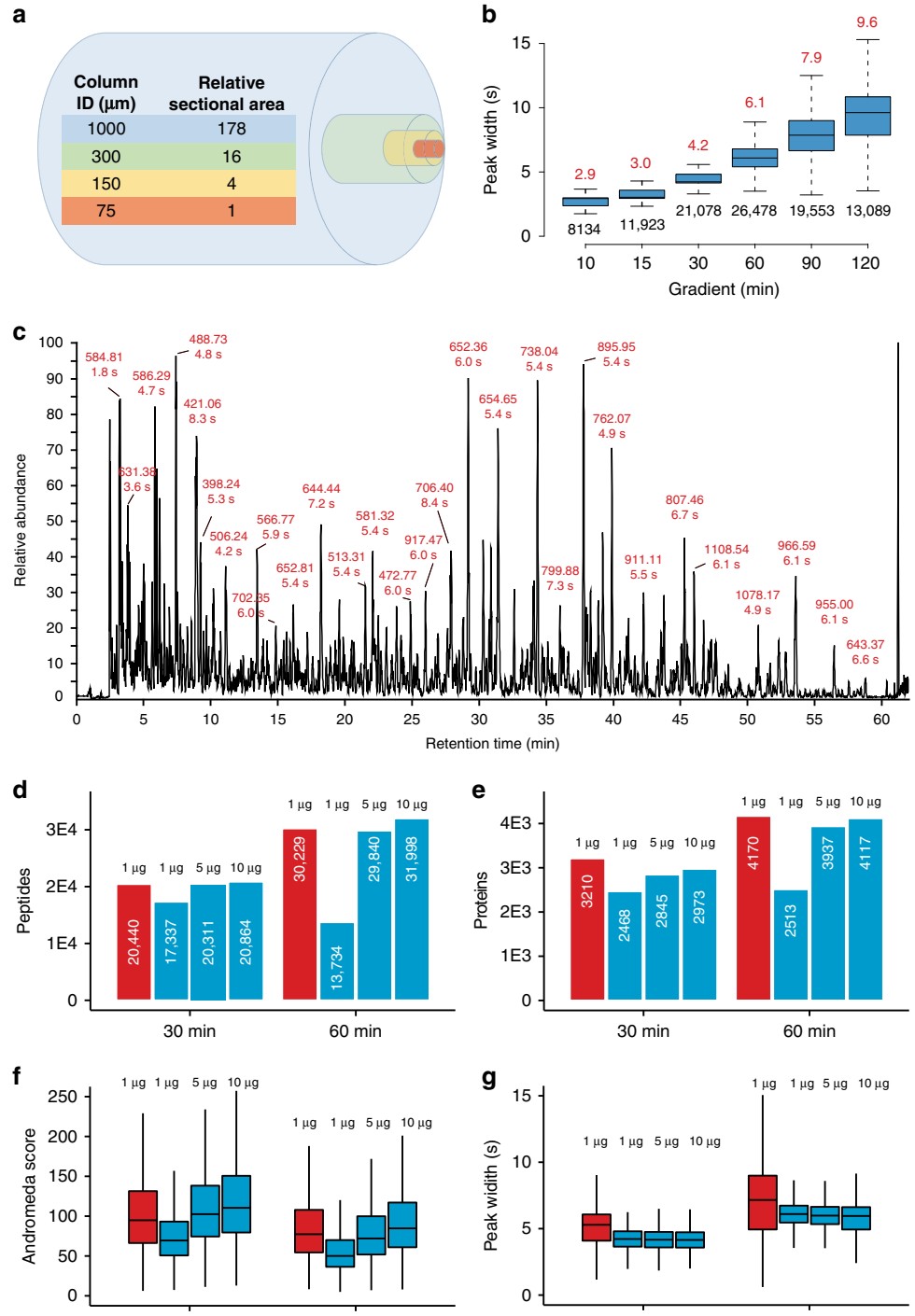

**Fig. 1 Qualitative performance characteristics of the micro-flow LC–MS/MS system. a** Comparison of the cross-sectional areas of LC columns of different inner diameters (ID). **b** Boxplots summarizing the chromatographic peak width distributions (full width at half-maximum, FWHM) of all identified peptides for different LC gradient times (2 μg HeLa protein digest injected). Boxes and whiskers cover 50% and 1.5 times the interquartile range of the data respectively. Numbers above boxes denote the median FWHM values (in seconds), numbers below boxes represent the number of peptides contained in the analysis. **c** Example base peak chromatogram of 2 μg HeLa protein digest separated by a 60 min LC gradient. Selected chromatographic peaks are labeled with the m/z and FWHM values of the underlying peptide. **d** Bar charts comparing peptide identification results obtained for different sample loadings and LC gradient times using either nano-flow[29, 31] (red) or micro-flow LC–MS/MS (blue). White numbers inside bars denote the number of peptides. **e** Same as panel **d** but for proteins. **f** Box plots comparing Andromeda peptide identification scores obtained for different sample loadings and LC gradient times using either nano-flow (red, $n = 24,113$ for 30 min, 36,442 for 60 min) or micro-flow (blue, $n = 19,514$ (1 μg), 23,047 (5 μg), and 23,630 (10 μg) for 30 min, and $n = 14,992$ (1 μg), 34,356 (5 μg), and 37,130 (10 μg) for 60 min) LC–MS/MS. **g** Same as panel **f** but for peptide chromatographic peak widths. The number of peaks used for each box are 22,254 (nano), 19,104 (micro, 1 μg), 21,820 (micro, 5 μg), and 22,065 (micro, 10 μg) for 30 min, and 33,772 (nano), 12,831 (micro, 1 μg), 32,835 (micro, 5 μg), and 34,884 (micro, 10 μg) for 60 min. Boxes and whiskers are defined as in **b**. Source data are provided as a Source Data file for **b**, **f**, and **g**.

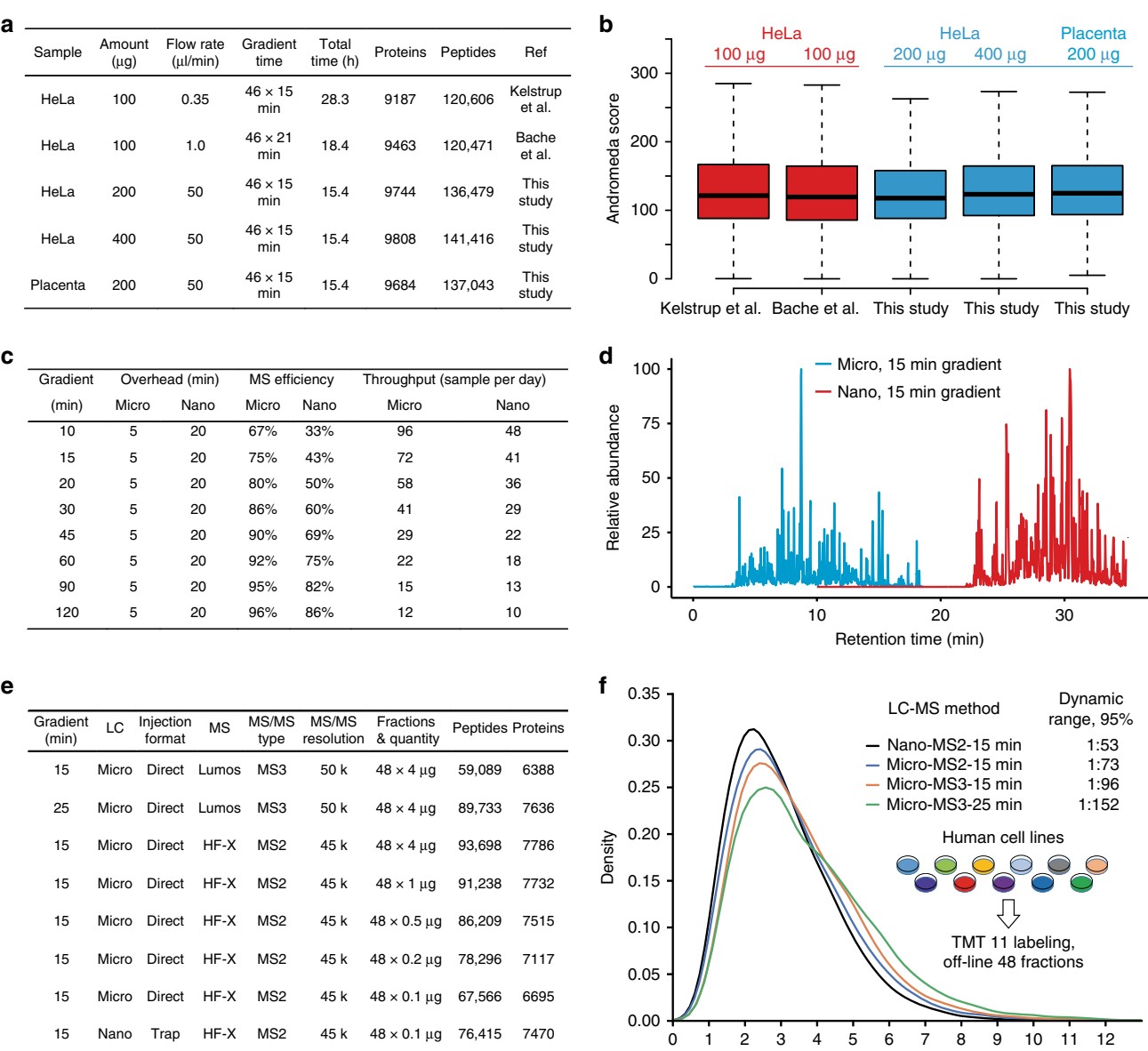

**Fig. 2 Deep-scale proteome analysis of the micro-flow LC–MS/MS system. a** Summary of key experimental paramters and results comparing deep-scale proteome analysis data of HeLa and placenta protein digests using published nano-flow LC–MS/MS data and data obtained by micro-flow LC–MS/MS data in this study. **b** Box plots comparing Andromeda peptide identification scores of the data shown in **a**. Boxes and whiskers are defined as in Fig. 1b. The number of peaks used for each box are 303,597 (Kelstrup et al.[31]), 282,215 (Bache et al.[29]), 287,834 (This study, 200 μg HeLa), 304,278 (This study, 400 μg HeLa), and 276,306 (This study, Placenta). Boxes and whiskers are defined as in Fig. 1b. Source data are provided as a Source Data file for **b**. **c** Summary of the actual sample throughput that can be achieved by the micro-flow LC–MS/MS system presented in this study compared to a typical nano-flow LC–MS/MS setup. **d** Overlay of base peak chromatograms of the same high pH reversed phase chromatography fraction of a TMT11 labeled pepties from a deep-scale analysis of 11 human cancer cell lines analyzed by micro-flow (blue) and nano-flow (red) LC–MS/MS, each using a 15 min gradient. **e** Result summary for the deep-scale proteomic analysis of eleven TMT-multiplexed human cancer cell lines using nano-flow or micro-flow LC–MS/MS systems. **f** Density plot displaying the dynamic range of TMT quantification for common peptides from 11 human cancer cell lines obtained by measuring identical samples using the LC–MS/MS configurations as shown in **e**. Dynamic range of TMT quantification was defined as the ratio of the maximum and minimum intensity values of a peptide across the 11 TMT channels.

LC–MS/MS time (Fig. 2e). The performance gap between the MS2 and MS3 measurements on the Lumos could be closed by extending the LC gradient time per fraction from 15 to 25 min leading to an increase of total analysis time from 16 to 24 h for the 11 proteomes. Again, the peptide quantities required to achieve this performance level were only 2–5 times higher than those for nano-flow LC (Fig. 2e). The improved chromatographic separation performance of the micro-flow system also led to an improved practical dynamic range of protein expression

quantification between the 11 human cell lines (Fig. 2f). Specifically, dynamic range (defined to cover 95% of all protein expression ratios between cell lines, see methods) increased from 1:50 (nano-flow) to 1:75 (micro-flow) for MS2 measurements, to 1:100 (micro-flow) for MS3 measurements all using 15 min gradients, and to 1:150 (micro-flow) when extending the gradient time from 15 to 25 min (Fig. 2f). These above results demonstrate that deep-scale proteome analysis of higher organisms can be envisaged at a throughput of 11–16 proteomes per day.

**Robustness and reproducibility of micro-flow LC–MS/MS.** To demonstrate robustness and to explore the quantitative performance of the micro-flow LC system, we set up an experiment consisting of 1550 consecutive injections organized into 10 identical cycles (or batches) of 155 injections each (Fig. 3a) that were analyzed over the course of ~40 days. In each cycle, we analyzed 20 replicates of 2 µg of HeLa, 5 µg of urine, 5 µg of cerebrospinal fluid (CSF) and 5 µg of plasma (4 replicates from 5 individuals) protein digests as well as one deep-scale human placenta protein digest using 200 µg starting material. In total 500 fmol of synthetic peptide retention time (RT) standards[32] (PROCAL, 40 peptides) were spiked into every single shot sample and 3 replicates of 500 fmol

PROCAL runs were added between each sample type to assess carry-over. We observed very stable peptide and protein identification numbers for the first 8 cycles, after which mass spectrometric performance dropped as a result of the accumulation of contaminants (Supplementary Figs. 5, 6 and Source Data File). In contrast, the performance of the micro-flow LC system was stable throughout (Fig. 3b and Source Data File, Supplementary Fig. 7 and Source Data File). The retention times (RT) of the spiked PROCAL peptides showed an average CV of 0.26% ($n > 40,000$ data points) demonstrating very high chromatographic reproducibility. Remarkably, the RTs of PROCAL peptides measured alone or spiked into body fluid samples were essentially the same (Pearson

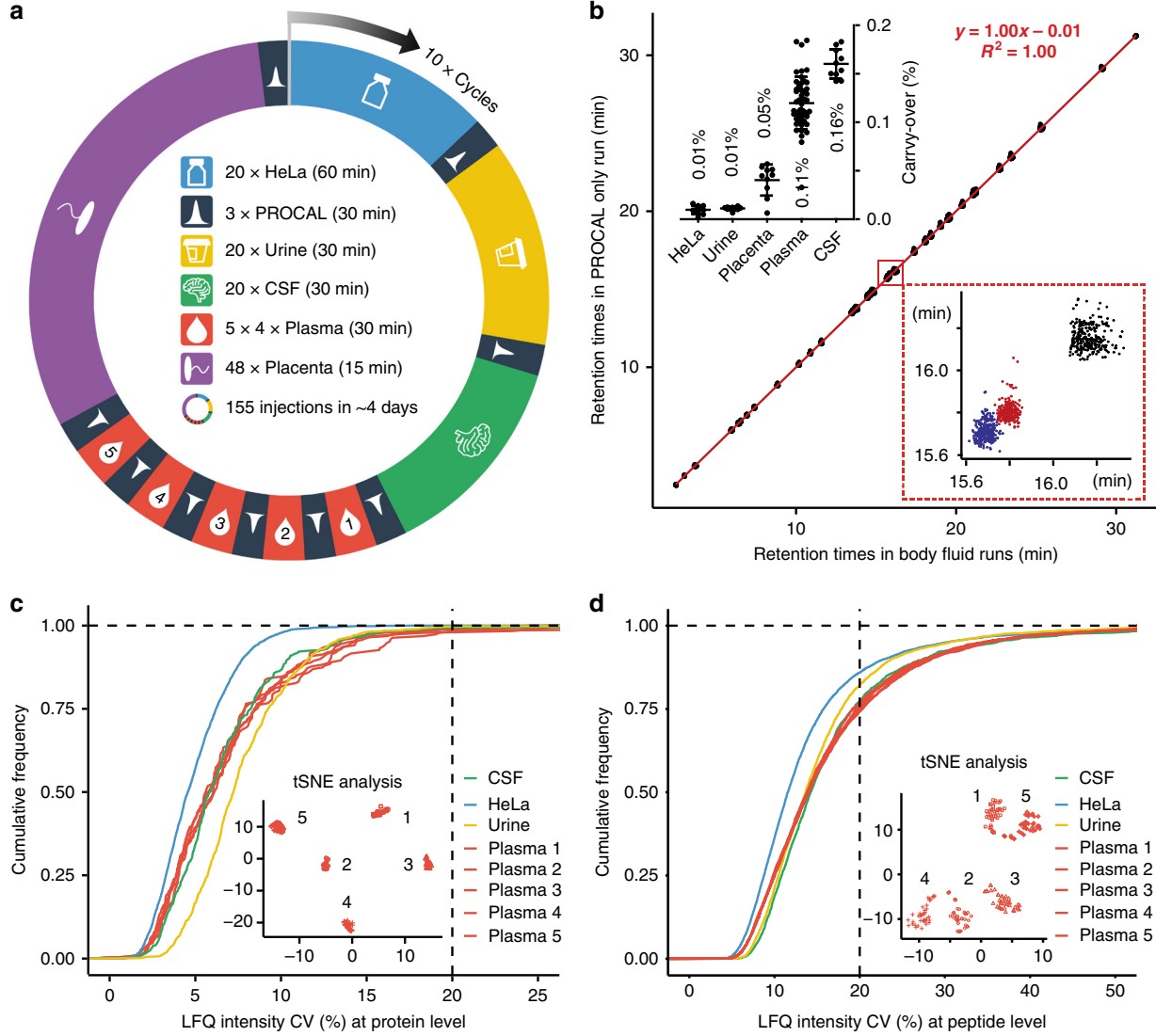

**Fig. 3 Quantitative performance characteristics of the micro-flow LC–MS/MS system. a** Design of a long-term performance test consisting of 10 cycles of 155 injections each. In each cycle, 20 replicates of 2 µg HeLa, 5 µg urine, 5 µg cerebrospinal fluid (CSF), and 5 µg each of 4 replicates of plasma digests from five individuals, as well as one deep-scale placenta digest (200 µg protein digest separted into 48 fractions) were analyzed using micro-flow LC–MS/MS using the stated gradient times. Between each sample type, 3 replicates of 500 fmol PROCAL synthetic retention time standards were injected. **b** Retention time stability of PROCAL peptides measured alone or spiked into body fluids (urine, CSF, plasma samples; main plot). The equation represents the linear model that was fitted to the data ($R^2$, squared Pearson correlation coefficient). The bottom right inset shows an expanded view of the main plot showing the retention time distribution of three closely eluting PROCAL peptides (in different colors) across all experiments. The top left inset summarizes the carry-over analysis across all 10 cycles (columns denote average carry-over and error bars denote the standard deviation). Source data are provided as a Source Data file. **c** Cumulative density plot showing the inter-cycle reproducibility of protein quantification for 200 injections of HeLa, urine and CSF, and 40 injections of five individual plasma samples (common proteins only). Dotted lines denotes the percentage of proteins that show <20% coefficient of variation (CV) in the analysis The inset shows a t-SNE analysis of the 40 plasma injections measured from each of five individuals across the 10 cycles. **d** Same as in **c** but at the peptide level.

$R^2$ of 1.00; slope of 1.00, intercept of <0.01 min), demonstrating the near absence of chromatographic matrix effects[33] (Fig. 3b and Source Data File). In addition, sample carry-over was extremely low (average of 0.16% for CSF, 0.11% for plasma, 0.05% for placenta and 0.01% for urine and HeLa; Fig. 3b and Source Data File) removing a common issue of nano-flow LC particularly for the analysis of tissues and body fluids[34]. The low carry-over is likely owing to the very low amount of sample loaded on the column relative to its capacity and the high volume of solvents passing over the column.

The exquisite chromatographic reproducibility also led to a very high reproducibility of protein and peptide quantification (Fig. 3c, d). Across all single shot samples, the median CV of common quantified proteins was between 4.6% (HeLa) and 7.2% (urine). Variation within cycles was even smaller (Supplementary Fig. 8) and practically all proteins had CVs below 20%. At the peptide level, the median CV of common quantified peptides was between 11.6% (HeLa) and 14.2% (CSF) and variation within cycles was again smaller (Supplementary Fig. 9). Between 75% and 85% of all quantified peptides showed CVs of <20% (Fig. 3d). Reproducibility of quantification was further assessed by a t-SNE analysis[35] at both protein (Fig. 3c) and peptide (Fig. 3d) level that clustered the 40 replicates of the 5 individuals from whom plasma was repetitively analyzed across the 10 cycles. Batch effects between the 10 cycles were observed for each sample type, but on a much smaller scale compared to the differences between human subjects (Supplementary Fig. 10).

The above figures of merit are very encouraging, as most quantified proteins would pass guidelines on quality specifications for clinical assays[36]. Such performance characteristics would be difficult if not impossible to achieve by nano-flow LC separations because of the very high dynamic range of protein concentrations present particularly in body fluids. Such high dynamic range compromises the quality of chromatographic separations at sample loadings that also yield high numbers of peptide and protein identifications. Both issues can be overcome using the micro-flow setup presented here. Using 30 min LC gradients, about 250 plasma proteins (2300 peptides), 600 CSF proteins (4500 peptides) and 1100 urine proteins (5000 peptides) were identified from 5 µg of protein digest (Supplementary Figs. 5, 6 and Source Data File, and Supplementary Fig. 11 and Source Data File). Such quantities are easily obtained from biological sources and the number of identifications is comparable to figures reported in the recent nano-flow LC literature[37–39]. In contrast to nano-flow LC, the micro-flow LC system showed no obvious sign of overloading even when injecting as much as 20 µg protein digest of non-depleted plasma protein digest, (Supplementary Fig. 12a). However, there is evidence that the mass spectrometric signal may saturate at high loading of plasma samples for some high abundant peaks (Supplementary Fig. 12b).

An further important consideration for high-throughput applications or experiments requiring reproducible results over extended periods of time is column lifetime. During the 40 days of the long-term performance test described above, the column showed very high separation reproducibility throughout as demonstrated by the overlay of the base peak chromatograms of 10 urine samples (one from each cycle; Supplementary Fig. 13a). At the time of writing, the column that was used in this study had separated >7500 samples over the course of about 1 year with no apparent loss of performance (Supplementary Fig. 13b and Source Data File) further attesting to the high potential of this set-up for implementation in clinical proteomics research.

**Sub-proteome analysis by micro-flow LC–MS/MS.** We extended the range of applications of micro-flow LC–MS/MS to the analysis of proteomes of lesser complexity, notably to protein-protein interactions using affinity purification (AP) of tagged proteins[40] or employing proximity labeling (BioID)[41]. Single-shot analysis of AP-MS and BioID-MS experiments (2 biological × 3 technical replicates for each bait protein) on the micro-flow LC system (15 min gradients each) recovered between 86% and 96% of the high-confident interactors identified in a previous publication[40] and in the Human Cell Map project[42] (Fig. 4a, Supplementary Data 1, 2). Both previous analysis used nano-flow LC systems and much longer gradient times. We also downloaded the ten most highly confident interaction partners of each bait from the String database[43] (Supplementary Data 3) and found that most of these interaction partners were also identified as high confidence interactors in the mirco-flow LC–MS/MS data and subsequent analysis by the software package SAINT[44,45] (Fig. 4b, c, Supplementary Fig. 14).

We next applied the micro-flow approach to the analysis of drug-protein interactions using the kinobeads approach[46]. Illustrated by the kinase inhibitor AT-9283 as an example (Fig. 4d), the micro-flow LC system covered >90% of the measured protein kinases but in less than a third of the gradient time compared to the same experiment performed by nano-flow LC (Supplementary Fig. 15a, b). More importantly, the main targets of the drug were all identified and the effective concentration of drug needed to compete 50% of the bound kinase ($EC_{50}$) obtained from the dose-response curves characterizing the interactions were very similar between experiments measured by micro- or nano-flow LC–MS/MS (Supplementary Fig. 15c–f).

As shown above (Fig. 1b), micro-flow LC separations using short gradient times generated very sharp LC peaks (here, median of 3.0 s, FWHM) and 8–10 data points across an LC peak are typically required for accurate determination of the LC peak area and thus quantification. Therefore, it is important to match the cycle time of the mass spectrometer (i.e. the time between two MS1 scans) to the chromatographic resolution (here ~0.6 s; see also Supplementary Fig. 2). Figure 4e shows extracted ion chromatograms for the AURKA peptide QWALEDFEIGRPLGK as an example and the data shows that sufficient data points cover the LC peak at all drug doses allowing the determination of the peak area with good confidence. This is even more important for the analysis of post-translational modifications such as phosphopeptides, as their quantification cannot be stabilized by aggregating several peptide quantification measurements into one protein quantification value. The dose response curves shown in Fig. 4f representing two peptides containing the phosphorylation site pS21 (ARTSpS-FAEPGGGGGGGGGGGSASGPGGTGGGK and TSpSFAEPGGG GGGGGGGSASGPGGTGGGK) and two peptides containing the site pY279 (QLVRGEPNVSpYICSR and RGEPNVSpYICSR) of the kinase GSK3A show that the $EC_{50}$ values obtained from the individual phospho-peptides are very similar to that of the aggregated protein indicating very good quantification also at the peptide level. Given such qualitative and quantitative performance levels, it becomes feasible to screen the targets of hundreds to possibly thousands of kinase inhibitors in this way.

**Analysis of phosphoproteomes by micro-flow LC–MS/MS.** Finally, we investigated the merits of the micro-flow LC system for phosphoproteome analysis. To this end, 2 mg of HeLa protein digest were fractionated by off-line high pH reverse phase chromatography into 96 fractions that were pooled into 12 fractions followed by IMAC phosphopeptide enrichment of each of the 12 fractions. Two workflow replicates were prepared and the phosphopeptides were analyzed by either micro-flow or nano-flow LC–MS/MS coupled to an Orbitrap QE HF-X mass spectrometer and using the same (60 min) LC gradient and MS data acquisition

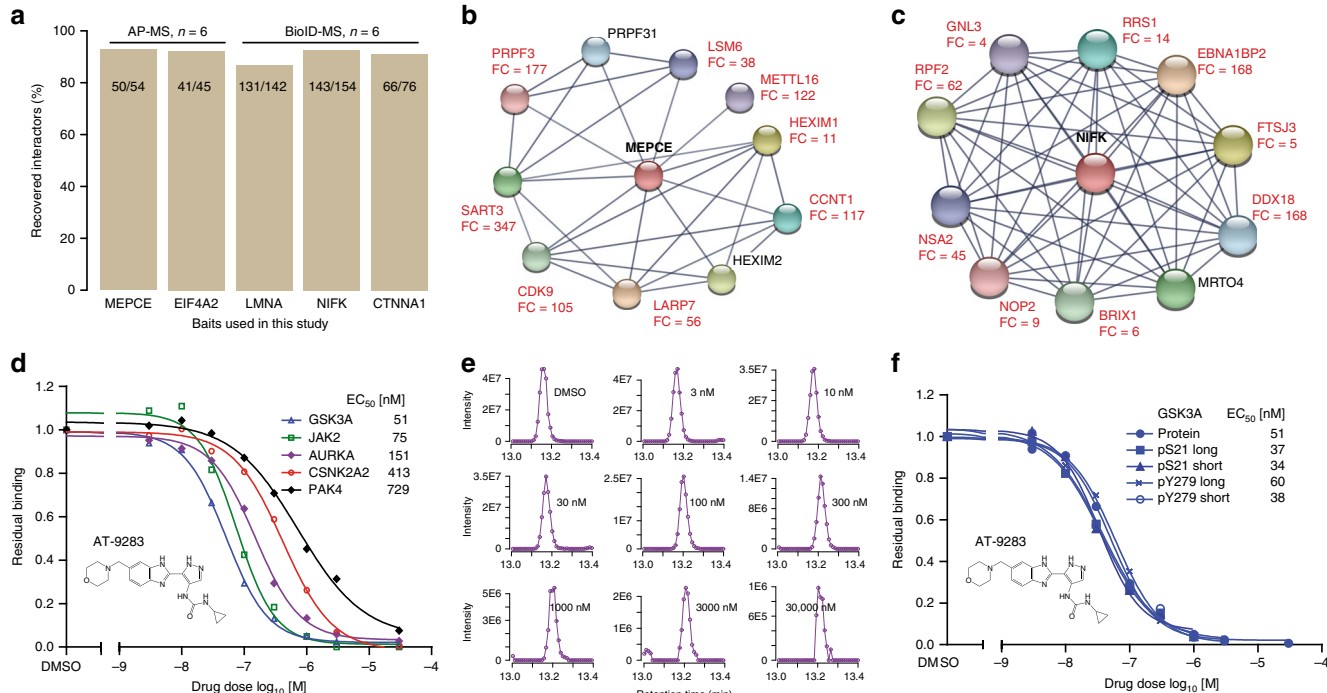

**Fig. 4 Application of the micro-flow LC–MS/MS system to the analysis sub-proteomes. a** Recovery analysis of high-confidence interactors obtained by replicate analysis ($n = 6$; two biological replicates and three technical replicates) of affinity purifications performed using FLAG-tagged human MEPCE and EIF4A2, and BioID proximity labeled human LMNA, NIFK, and CTNNA1 and analyzed by one-shot micro-flow LC–MS/MS using 15 min gradients (AP-MS and BioID-MS), compared to results of the same experiments published previously or part of the Human Cell Map project. Numbers inside the bars represent the number of interactors identified by micro-flow LC–MS/MS vs those annotated in the aforementioned resources. **b** Interaction networks based on the ten most confident interaction partners in STRING for the bait proteins MEPCE (FLAG affinity purification). FC denotes the fold change values of an interaction partner (over control pulldowns) assessed by SAINTexpress analysis. Proteins without FC annotations indicate that they were not identified as high confident interactors in this micro-flow LC–MS/MS study. **c** Same as **b** but for NIFK (BioID proximity labeled). **d** Example dose-response curves of protein kinases that are the targets of the kinases inhibitor AT-9823 obtained by kinobeads competition pulldown experiments and analyzed by micro-flow LC–MS/MS system using a 15 min gradient for each drug dose. EC$_{50}$ values (effective concentration 50) denote the drug concentration necessary to compete 50% of the binding of a protein to kinobeads. **e** Extracted ion chromatograms of the AURKA peptide QWALEDFEIGRPLGK from the kinobead experiment in **d** illustrating the quantification of this peptide as a function of the applied drug dose. **f** Dose–response curves akin to **d** but for four phosphopeptides containing the phosphorylation sites pS21 and pY279 of the protein kinase GSK3A, as well as the aggregated data for the entire protein.

methods. Somewhat surprisingly given the overall low abundance of phosphopeptides (typically ~1% relative to the total), micro LC–MS/MS identified 32,493 unique phosphopeptides and 27,639 phosphorylation sites corresponding to 4,886 phosphoproteins within 12 h of gradient time. The same sample analyzed by nano-flow LC resulted in the identification of 28%, 14%, and 18% more phosphopeptides, phosphorylation sites and phosphoproteins respectively, confirming that nano-flow LC plays out its advantages when sample quantities are low (Fig. 5a–c). This is also reflected by a lower identification score (Fig. 5d and Source Data File) as a result of the lower absolute signal intensity in micro-flow LC–MS/MS compared to nano-flow LC–MS/MS. As expected, LC peaks were sharper for micro-flow LC compared to nano-flow LC separations (Fig. 5e and Source Data File) which may improve the separation of phosphorylation site isomers. Using the peptide SGAQASSTPLSPTR of Lamin A/C as an example, Fig. 5f shows that separation of multiple such singly phosphorylated peptide isomers is indeed possible (for further examples, see Supplementary Fig. 16). Also, the micro-flow LC system is more efficient in separating the S18 and S22 phosphorylation isomers as deduced from the ~40% higher ratio of the difference in retention time (deltaRT) divided by the chromatographic peak width. From our data, it is, however, not clear if this higher separation efficiency generally translates into a clear advantage for the micro-flow LC setup for the separation of phosphorylation isomers as we did not find enough cases of

closely eluting phosphopeptide isomers on which this hypothesis could be tested. Nevertheless, it is noteworthy that the micro-flow LC setup performed quite well for the analysis of phosphoproteomes, which makes this approach worth considering for high-throughput applications or for projects in which sample availability is not a concern.

In conclusion, this study showed that micro-flow LC–MS/MS is a very versatile alternative to the conventional nano-flow LC approach for a broad range of proteomic applications. Because of its robust qualitative and quantitative performance characteristics, the simplicity of implementation and the very broad range of available high-quality micro-flow columns, the authors expect that the approach will be broadly enabling for many expert as well as non-specialized laboratories. The approach also paves the way for more routinely translating proteomics into clinical applications, particularly for quantitative and high-throughput body fluid analysis. In addition, the very large amount of data collected in this project, should be a useful resource for the scientific community to further investigate aspects of the methodology that are not covered in this report.

## Methods

**Sample selection and preparation.** Human specimen used in this study were obtained following informed consent and observing the appropriate ethics approval process of the Technical University of Munich. The study was approved by the ethics committee of the faculty of medicine of the Technical University of Munich.

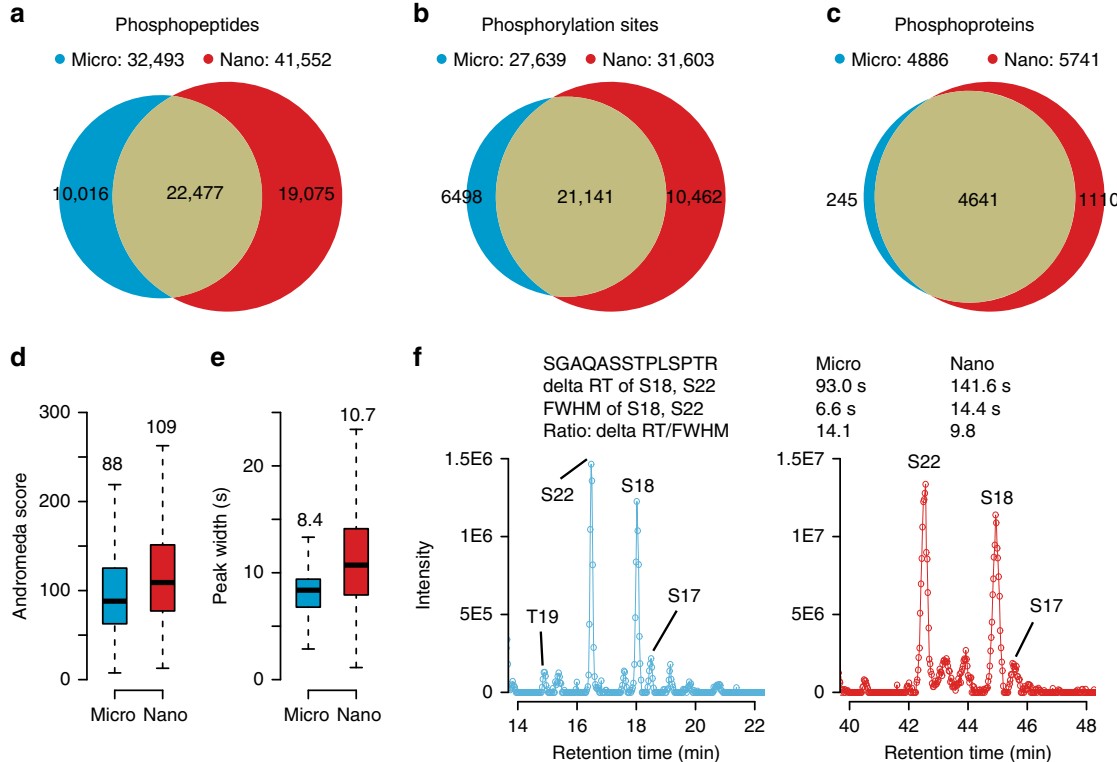

**Fig. 5 Micro vs nano-flow LC–MS/MS for analysis of phosphoproteomes. a** Venn diagram showing the number and overlap of phosphopeptides identified by nano-flow LC–MS/MS (red) and micro-flow LC–MS/MS (blue) using 2 mg of HeLa protein digest, separated into 12 high-pH reversed phase chromatography fractions, enriched for phosphopeptides using IMAC and analyzed by 60 min LC gradients. **b** Same as in panel **a** but for phosphorylation sites. **c** Same as in **a** but for phosphoproteins. **d** Boxplots showing the Andromeda score distribution for peptides identified by nano-flow LC–MS/MS (red, $n = 109,417$) and micro-flow LC–MS/MS (blue, $n = 77,163$). Boxes and whiskers are defined as in Fig. 1b. Numbers above box plots denote the median Andromeda score. **e** Same as in **d** but for the chromatographic peak width of the peptides (red, $n = 102,066$, blue, $n = 83,908$). Source data are provided as a Source Data file for **d** and **e**. **f** Extracted ion chromatograms of phosphorylation site isomers of the peptide SGAQASSTPLSPTR from human lamin A/C identified by micro-flow LC–MS/MS (blue) or nano-flow (red) LC–MS/MS. deltaRT refers to the difference in retention time between the S18 and S22 phosphorylation isomers. FWHM refers to the full chromatographic peak width at half maximum of the two phosphopeptides and ratio: deltaRT/FWHM refers to the ratio of the two values.

In total 1 mL plasma samples collected from five healthy donors were centrifuged for 10 min at 4000 g. In total 50 µl supernatant were taken out and diluted by 5 volumes of 8 M urea buffer containing 80 mM Tris–HCl, pH = 7.6, and then stored at −80 °C until further use. Second-morning mid-stream urine was collected from a healthy donor, cooled immediately to 4 °C, and centrifuged at $4000 \times g$ at 4 °C for 30 min to remove cell debris. The supernatant was vacuum concentrated 5-fold using a SpeedVac and proteins were precipitated using five volumes of ice-cold ethanol, incubated at −20 °C overnight, and followed by centrifugation at $20,000 \times g$, 4 °C for 30 min. The pellet was dissolved in 8 M urea buffer containing 80 mM Tris–HCl, pH = 7.6, and stored at −80 °C until further use. The cerebrospinal fluid (CSF) samples from five individuals with relapsing/remitting multiple sclerosis (RRMS) and five healthy individuals were pooled to obtain sufficient sample for this technical study. The pooled CSF was stored at −80 °C until further use. Because the protein concentration in the CSF sample was very low, no urea buffer was used. About 1 g fresh frozen human placenta tissue was washed five times with ice-cold PBS buffer containing protease inhibitors, followed by addition of 3 mL lysis buffer containing 8 M Urea, 80 mM Tris–HCl (pH = 7.6), 1 × EDTA-free protease inhibitors (complete mini, Roche). The lysate was transferred to precellys tubes containing ceramic beads and incubated on ice for 5 min. Then, the precellys tubes were mounted on a Precellys 24 bead-milling device (Bertin Instruments). Tissue homogenization was performed at 5500 rpm for $2 \times 25$ s with 5 s pause for five cycles. Between each cycle, the lysate was put on ice for 10 min. The lysate was centrifuged at 20,000 g, 4 °C for 30 min, the supernatant was stored at −80 °C until further use. Human epithelial cervix carcinoma HeLa cells (ATCC, CCL-2) were cultured in DMEM (Gibco, Invitrogen), supplemented with 10% fetal bovine serum, 100 U/mL penicillin (Invitrogen), 100 µg/mL streptomycin (Invitrogen), at 37 °C, in a humidified incubator with 5% CO2. Cells were harvested at ~80% confluence by washing twice with PBS buffer and subsequently adding lysis buffer containing 8 M Urea, 80 mM Tris–HCl (pH = 7.6), 1 × EDTA free protease inhibitors (complete mini, Roche), and 1 × Phosphatase inhibitors (Sigma Aldrich) directly to the cell culture plate. The plate was incubated on ice for 5 min. The cell lysate was collected by scraping the plate. After that, the lysate was centrifuged at

$20,000 \times g$, 4 °C for 30 min. The resulting supernatant was stored at −80 °C until further use. The 10 pancreatic cell lines (BxPC-3 (ATCC, CRL-1687), Dan-G (DSMZ, ACC 249), HPAC (ATCC, CRL-2119), HuPt-4 (DSMZ, ACC 223), IMIM-PC-1 (PRID, CVCL_4061), MiaPaCa2 (ATCC, CRM-CRL-1420), Panc-10.05 (ATCC, CRL-2547), Pa-Tu-8998-S (DSMZ, ACC 204), Pa-Tu-8998-T (DSMZ, ACC 162) and PSN-1 (ATCC, CRL-3211)) were provided by Günter Schneider. The cell lines were cultured according to the cell line provider's recommendations to 60–80% confluency[47]. The cells were lysed with 8 M urea, 40 mM Tris/HCl (pH 7.6), 1 × EDTA-free protease inhibitor mixture (Complete Mini, Roche), and 1 × phosphatase inhibitor mixture (Sigma-Aldrich). The cell lysate was clarified by centrifuged at $20,000 \times g$ for 20 min. The supernatants were used for in solution trypsin digestion.

**Protein digestion and peptide desalting.** Protein concentration was measured by the Bradford assay. Proteins were reduced by 10 mM DTT at 37 °C for 60 min, and alkylated by 55 mM chloroacetamide (CAA) at room temperature for 30 min in the dark. For the CSF sample, the protein solution was mixed with one volume of 40 mM Tris–HCl (pH = 7.6). For all the other samples, the protein solution were mixed with five volumes of 40 mM Tris–HCl (pH = 7.6). Proteins were digested with sequencing grade trypsin (Roche) at a protease-to-protein ratio of 1:100 (w/w) for 4 h, followed by adding further trypsin (1:100) and incubating overnight at 37 °C. Digestion was quenched by addition of formic acid (FA) to a final concentration of ~1%, and the resulting peptide mixture was centrifuged at $5000 \times g$ for 15 min. Peptides in the supernatant were loaded on Sep-Pak C18 Cartridges (Waters) and eluted by 50% ACN, 0.1% FA in water and dried in a SpeedVac. Samples were stored at −80 °C until further use.

**TMT labeling.** For TMT11-plex labeling, desalted peptides from HeLa, and ten pancreatic cell line protein digests were reconstituted in 0.1% FA and peptide concentration was determined by NanoDrop[TM] 2000 (Thermo Scientific). The peptides from the cell lines of MiaPaCa2, HuPt-4, BxPC-3, HPAC, Panc-10.05,

PSN-1, Dan-G, Pa-Tu-8998-S, Pa-Tu-8998-T, IMIM-PC-1 and HeLa were labeled with the 126, 127 N, 127 C, 128 N, 128 C, 129 N, 129 C, 130 N, 130 C, 131 N, and 131 C of the TMT11 reagent. TMT labeling was performed according to the our published protocol[48], Briefly, 200 µg peptides of each of the eleven cell lines were dried in a SpeedVac and reconstituted in 40 µl of 50 mM HEPES buffer (pH 8.5). Then, 0.2 mg TMT reagent in 10 µl dry ACN was added to each sample and mixed with a pipette. The mixture was incubated at 25 °C and 400 rpm on a thermomixer for 1 h. After that, 4 µl of 5% Hydroxylamine solution was added to each sample to stop the reaction, the reaction was performed at 25 °C and 400 rpm on a thermomixer for 15 min. Finally, the labeled peptides were pooled together and 40 µl of 10% FA in 10% ACN were added. To avoid one SpeedVac step before peptide desalting, the pooled peptides were diluted by 20 volumes of 0.1% FA, and purified by the Sep-Pak C18 Cartridge and the eluate was dried in a SpeedVac and stored at −80 °C until further use.

**Off-line high pH reversed phase peptide fractionation**. A Dionex Ultra 3000 HPLC system operating a Waters XBridge BEH130 C18 3.5 µm 2.1 × 250 mm column was used to fractionate peptides at a flow rate of 200 µl/min. Buffer A was 25 mM ammonium bicarbonate (pH = 8.0), buffer C was 100% ultrapure water (ELGA), buffer D was 100% ACN, buffer B was not used in this system. The proportion of buffer A was kept at 10% during separation. Fraction were collected every minute and fractions were collected into a 96 well plate. For non-labeled peptides, the 200 or 400 µg protein digests were separated by a linear gradient from 5% D to 30% D in 87 min, and followed by a linear gradient from 30% D to 80% D in 5 min.

To be able to compare results obtained in this study to data from the literature, 92 fractions were collected (2 min to 94 min) and subsequently pooled into 46 fractions by adding fraction 47 to fraction 1, fraction 48 to fraction 2 and so forth. The HeLa protein digest (200 µg and 400 µg) and placenta protein digest (200 µg) were fractionated into 46 fractions to compare with the published data. For the placenta digest used for the 10 cycles' long-term test, we collected 96 one minute fractions (between 1 min and 97 min) and pooled these into 48 fractions (as above). For TMT labeled peptides, 500 µg pooled peptides were separated by a linear gradient from 9% D to 42% D in 86 min, followed by a linear gradient from 42% D to 80% D in 12 min. 96 fractions were collected and pooled into 48 fractions (as above). Peptide fractions were frozen at −80 °C freezer for at least 1 h and dried in a SpeedVac without prior desalting.

**Phosphopeptide separation and Fe-IMAC enrichment**. In total 2 mg HeLa protein digest was separated on a 2.1 × 150 mm Waters XBridge BEH130 C18 3.5 µm column with a linear gradient from 4% D to 32% D in 45 min, ramped to 80% D in 6 min, and kept there for 3 min before ramped back to 5% D in 2 min and 96 fractions were collected at 0.5 min intervals. Peptides were pooled in a stepwise fashion from 96–48 to 24–12 fractions akin to the scheme above. Fractions were dried in a SpeedVac and stored at −80 °C until performing phosphopeptide enrichment. Phosphopeptides were enriched from each of the 12 fractions using Fe (III)-IMAC-NTA (Agilent Technologies) on the AssayMAP Bravo Platform (Agilent Technologies). IMAC cartridges were primed with 100 µl of 99.9% ACN/0.1% TFA and equilibrated with 50 µl loading buffer (80% ACN/0.1% TFA). Samples were dissolved in 200 µl of loading buffer and loaded onto cartridges. The cartridges were washed with 50 µl loading buffer, and phosphopeptides were eluted with 40 µl of 1% ammonia. Phosphopeptides were dried down and stored at −80 °C until subjected to LC–MS/MS analysis.

**Kinobeads, FLAG based APs and BioID pull-downs**. Kinobeads selectivity profiling of AT-9283 was performed with the standard protocol[49]. The K-562 (ATCC, CCL-243), COLO-205 (ATCC, CCL-222) and MV-4-11 (ATCC, CRL-9591) cells were cultured in RPMI 1640 medium (Biochrom GmbH) supplemented with 10% (v/v) FBS (Biochrom GmbH) and 1% (v/v) antibiotics. SK-N-BE(2) (ATCC, CRL-2271) cells were grown in DMEM/Ham's F-12 (1:1) supplemented with 10% or 15% (v/v) FBS, respectively and 1% (v/v) antibiotics (Sigma). OVCAR-8 (RRID: CVCL_1629) cells were cultured in IMDM medium (Biochrom GmbH) supplemented with 10% (v/v) FBS. Cells were lysed in 0.8% NP40, 50 mM Tris-HCl pH 7.5, 5% glycerol, 1.5 mM MgCl₂, 150 mM NaCl, 1 mM Na₃VO₄, 25 mM NaF, 1 mM DTT, protease inhibitors (SigmaFast, Sigma) and phosphatase inhibitors (prepared in-house according to Phosphatase inhibitor cocktail 1, 2, and 3 from SigmaAldrich).2.5 mg of a protein mixture from the five cell lines were pre-incubated with increasing concentrations of compound (DMSO vehicle, 3 nM, 10 nM, 30 nM, 100 nM, 300 nM, 1 µM, 3 µM, 30 µM) for 45 min at 4 °C in an end-over-end shaker. Subsequently, lysates were incubated with Kinobeads (18 µl settled beads) for 30 min at 4 °C on an end-over-end shaker. After washing, bound proteins were reduced with 50 mM DTT in 8 M Urea, 40 mM Tris HCl (pH = 7.4) for 30 min at room temperature. After alkylation with 55 mM CAA proteins were digested with trypsin. Peptides were desalted and concentrated using SepPak tC18 µEluation plates (Waters) and dried down in a SpeedVac. The affinity purification (AP) and BioID samples were prepared according to the previous protocol[50,51]. Briefly, the AP and BioID samples were prepared from HEK293 Flp-In T-REx lines inducibly expressing a test "bait" or a control that were induced for 24 h with 1 µM tetracycline or concomitantly with tetracycline and 50 µM biotin, and harvested on ice.

For BioID, cell pellets were resuspended in a 1:10 cell:buffer ratio. The lysis of cell pellets for BioID purification (on streptavidin-sepharose) was performed in 50 mM Tris-HCl (pH 7.5), 150 mM NaCl, 0.1% (w/v) SDS, 1% NP-40, 1 mM MgCl₂ and 1 mM EDTA, freshly supplemented with 0.5% final (w/v) sodium deoxycholate and protease inhibitors (Cat# P8340, Sigma-Aldrich). The suspension was sonicated 3 times for 5 s (2 s off) at 30% amplitude. TurboNuclease (BioVision Cat#9207-50KU; 250U/µl) and RNAse (BioBasic Cat#RB0474; 10 µg/µl) were added at 1 µl/sample and incubated at 4 °C for 30 min. In total 20% SDS was added to the sample such that the final concentration of SDS was 0.25%. This suspension was mixed well and centrifuged at 14,000 × g for 20 min at 4 °C. Supernatant was removed to a new tube containing 30 µl bed volume of streptavidin sepharose beads and incubated at 4 °C for 3 h with rotating. Cell pellets for FLAG APs were resuspended in a 1:4 cell:buffer ratio. FLAG affinity purification (on magnetic M2-antibody conjugated anti-Flag beads, Sigma-Aldrich, M8823) was performed for 2 h at 4 °C on a nutator[50,51]. After that, two washes with 1 mL of lysis buffer were performed and followed by an additional wash with 1 mL of 20 mM TrisHCl, pH 8, and 2 mM CaCl₂. Finally, samples were trypsinized on-beads overnight at 37 °C with rotating and without alkylation/reduction, and dried in a SpeedVac.

**Setup of the online micro-flow LC–MS/MS system**. A Dionex UltiMate 3000 RSLCnano System was coupled online to a Q Exactive HF-X or an Orbitrap Fusion Lumos mass spectrometer (Thermo Fisher Scientific) in this study. For the final LC setup, we used nanoViper capillaries for all the connections. The pump outlet was directly connected to sample injection valve (connected to one end of the sample loop) by a 75 µm ID × 550 mm nanoViper capillary, another 75 µm ID × 550 mm nanoViper capillary was used to connect the sample injection valve (connected to the other end of the sample loop) to the column inlet. A 20 uL sample loop was used with the micro-flow LC system in direct injection mode. The sample loop was kept in line during gradient elution. A 50 µm ID × 350 mm nanoViper capillary was used to connect the column outlet to the ground metal union of the Ion Max API source. Another 50 µm ID × 150 mm nanoViper capillary was used to connect the other end of the ground metal union to the sample inlet of HESI-II probe (50 µm ID) of the Ion Max API Source. The probe depth was set to A line of the Ion Max API source.

The results reported in this study were all obtained on a commercially available Thermo Fisher Scientific Acclaim PepMap 100 C18 LC column (2 µm particle size, 1 mm ID × 150 mm; catalog number 164711). Column temperature was maintained at 55 °C using the integrated column oven As a side note, we evaluated five batches of PepMap columns, which were produced in 2012, 2013, 2017, and 2018, respectively, and found that the 2012 and 2013 batch showed better separation efficiency, which is why the 2013 batch column was used throughout this study.

Three LC pumps available on Dionex UltiMate 3000 RSLCnano System (the loading pump, the NC gradient pump and a modified Vanquish pump) were used to deliver the gradient. The micro-flow LC–MS/MS system was initially developed by delivering gradients using the loading pump. The DMSO titration experiment was performed using the loading pump at a flow rate of 68 µl/min, and using linear gradients of 5–28% B, 4–27% B, 3–26% B, 2–25% B, 1–24% B, 0.1–23.1% B for solvents spiked with 0%, 1%, 2%, 3%, 4% and 5% DMSO, respectively. Although the loading pump can be used, we observed a rather long gradient delay, resulting from gradient mixing before the loading pump head and resulting in about 220 µl dead volume. Such gradient delays are unacceptably long for 15 min and 30 min gradients. Therefore, we installed a micro flowmeter (catalog number 6041.7903 A, maximum flow rate 50 µl/min) on the NC pump to deliver the gradient at the maximum flow rate of 50 µl/min and using a linear gradient of 3–28% B and including 3% DMSO in the solvents[19]. This NC pump setup was used for the method development parts of the manuscript, including gradient tests, serial HeLa dilution tests, deep-scale fractionated HeLa and placenta protein digests tests, etc. The presented data demonstrates that the NC pump/micro flowmeter setup was very robust.

However, as the highest flow rate for the NC pump/micro flowmeter combination is 50 µl/min, LC overhead time was still not optimal and should be improved particularly for short gradients. Therefore, a modified Vanquish pump capable of delivering gradients of up to 100 µl/min was used.

This modified Vanquish pump is a binary gradient pump, and has technical characteristics similar to standard high-pressure binary gradient pump in the NCS-3500RS module (https://assets.thermofisher.com/TFS-Assets/CMD/Specification-Sheets/PS-71899-LC-UltiMate-3000-RSLCnano-PS71899-EN.pdf) with a pump delay volume of <25 nL and a maximum pressure of 800 bar. There was no additional mixer installed between the pump outlet and fluidics. All observed delays in elution are associated only with the volume of the column, the injection loop and the capillary connections between pump, column, auto-sampler, injection loop, and HESI probe.

This makes it possible to flush the column at a flow rate of 100 µl/min and decrease the total overhead time to 5 min, including 3 min sample injection and 2 min column equilibration time. A flow rate of 50 µl/min was used to deliver the linear gradients. This setup enabled a throughput of 96 samples per day using 10 min gradient time. The modified Vanquish pump was used for the long-term performance test encompassing 1550 injections of different samples, the deep-scale

TMT labeled peptides, phosphopeptides, kinobeads pulldown samples, AP and BioID pull-down samples.

The final LC conditions we recommend for the micro-flow LC system described in this study are: solvent A: 0.1% FA, 3% DMSO in water; solvent B: 0.1% FA, 3% DMSO in ACN; direct sample injection using solvent A to load sample. A linear gradient of 3–28% B for all gradient lengths was used for the unlabeled full proteome analysis. Briefly, a linear gradient of 3–28% B in 15 min was used for deep-scale fractionated unlabeled peptides and all the pulldown samples (including kinobeads, AP and BioID pulldowns). In the long-term performance test, a linear gradient of 3–28% B in 30 min was used for urine, CSF, plasma and PROCAL peptides, a linear gradient of 3–28% B in 60 min was used for HeLa peptides, and a linear gradient of 3–28% B in 15 min was used for deep fractionated placenta peptides. For the enriched phosphopeptides, a linear gradient of 1–25% B in 60 min was used. For the deep-scale fractionated TMT labeled peptides, the linear gradients of 7–32% B in 15 min, 6–33% B in 25 min were used.

For micro-flow LC connected with Q Exactive HF-X, the Q Exactive HF-X mass spectrometer (Thermo Fisher Scientific) was operated as follows: Positive polarity; spray voltage 3.5 kV, funnel RF lens value at 40, capillary temperature of 320 °C, auxillary gas heater temperature of 300 °C. The flow rates for sheath gas, aux gas and sweep gas were set to 32, 5, and 0, respectively. Except otherwise noted, data dependent acquisition (DDA) using the Full MS-ddMS$^2$ setup was used. Full MS resolution was set to 60,000 at m/z 200 and full MS AGC target was 3E6 with a maximum injection time (IT) of 50 ms. Mass range was set to 360–1300. AGC target value for fragment spectra was set to 1E5. For MS2 spectra, the minimum AGC target was kept at 2E3. The isolation width was set to 1.3 m/z, and the first mass was fixed at 100 m/z. The normalized collision energy was set to 28%. Peptide match was set to preferred, and isotope exclusion was on. MS1 and MS2 spectra were acquired in profile and centroid mode, respectively. Further details of the respective MS methods are shown and discussed in the manuscript. Dynamic exclusion values were set to 10 s, 10 s, 15 s, 25 s, 40 s and 50 s for 10 min, 15 min, 30 min, 60 min, 90 min and 120 min gradients, respectively. For the full proteome analysis, the optimized 28 and 41 Hz methods were used, which were optimized in the previous literature[31]. The top N values in the 28 Hz and 41 Hz methods are 12 and 18, the MS2 spectra resolutions in the 28 Hz and 41 Hz methods are 15,000 and 7500, the maximum IT values of precursors in the 28 Hz and 41 Hz methods are 22 ms and 11 ms. The 41 Hz method was only used for the dilution test of the HeLa protein digest with 30 min and 60 min gradients. As we found 28 Hz method performed much better in most cases, the 28 Hz method was used for all the other full proteome samples including gradient test of HeLa protein digest, body fluid samples, deep fractionated HeLa and placenta protein digest, kinobeads, AP and BioID pulldown samples, etc. For enriched phosphopeptides, up to 12 precursors per cycle were picked for MS2 using a maximum IT of 120 ms and fragments were recorded at 15,000 resolution. The minimum AGC target was set to 5E3, all other parameters were kept the same. For the TMT labeled peptides, the AGC target value for fragment spectra was set to 2E5, up to 12 precursors per cycle were picked for MS2 and fragments were recorded at 45,000 resolution (maximum IT of 86 ms). The isolation window and normalized collision energy was set at 0.8 m/z and 35, the minimum AGC target was kept at 5E3.

For the micro-flow LC connected to the Orbitrap Fusion Lumos, the mass spectrometer (Thermo Fisher Scientific) was operated as follows: positive polarity; spray voltage 3.5 kV, capillary temperature 325 °C; vaporizer temperature 125 °C. The flow rates of sheath gas, aux gas and sweep gas were set to 32, 5, and 0, respectively. For the unlabeled peptides, full MS resolution was set to 120,000 at m/z 200, full MS AGC target was 4E5 with a maximum IT of 50 ms and RF lens value was set to 40. The mass range was set to 360–1300, and the MIPS properties were set to peptide. For MS2 spectra, the intensity threshold was set to 5E3, the default charges were set to state 2–6. The AGC target value was set to 1E4, isolation width was set to 0.4 m/z, and the first mass was fixed at 100 m/z. The ionTrap was used to detect MS2 spectra using the rapid scan function. The maximum IT was 10 ms for 15 min gradient, and 35 ms for 30 min gradient. The cycle time was set to 0.6 s for 15 min gradient, and 1 s for 30 min gradient. The dynamic exclusion duration was set to 12 s for 15 min gradient, and 20 s for 30 min gradient, exclude after one time. For the TMT labeled peptides, and the cycle time was set to 1.2 s for both 15 min and 25 min gradients. The dynamic exclusion duration was set to 40 s for 15 min gradient, and 50 s for 25 min gradient, exclude after one time. Full MS resolution was set to 60,000 at m/z 200 and full MS AGC target value was 4E5 with a maximum IT of 50 ms and RF lens value was set to 50. The mass range was set to 360–1500, and the MIPS properties were set to peptide. For MS2 spectra, the intensity threshold was set to 1E4, default charges were set to state 2–6. The ddMS2 IT HCD model was used for MS2 spectra, the isolation width was set to 0.6 m/z, activation type was HCD, HCD collision energy [%] was 32. The AGC target value was set to 1E4, and the first mass was fixed at 100 m/z. The ionTrap was used to detect the MS2 spectra using the rapid scan function. The maximum IT was 15 ms for 15 min gradient, 40 ms for 25 min gradient. The precursor selection range was set to 400–2000, exclusion mass widths were set to 20 m/z for low and 5 m/z for high. Isobaric tag loss exclusion properties were set to TMT reagent. The ddMS3 OT HCD model was used for MS3 spectra. Synchronous precursor selection was enabled, the number of SPS precursors was set to 8, the MS isolation window was 1.2 m/z, activation type was HCD, and HCD collision energy was 55%. The Orbitrap was used to detect the MS3 spectra at 50,000 resolution and over a scan range of 100–1000. The AGC target was 1E5 with a maximum IT of 86 ms for both 15 min and 25 min gradients.

**Nano flow LC-MS/MS.** A Dionex UltiMate 3000 RSLCnano system was coupled to a Q Exactive HF-X or Q Exactive HF mass spectrometer. Peptides were loaded onto a trap column (100 μm × 2 cm, packed in house with Reprosil-Gold C18 ODS-3 5 μm resin, Dr. Maisch) with solvent A (0.1% formic acid in HPLC grade water) at a flow rate of 5 μl/min for 10 min, and separated on an analytical column (75 μm × 40 cm, packed in house with Reprosil-Gold C18 3 μm resin, Dr. Maisch) at 300 nl/min. The analytical column was heated to 50 °C using a 30 cm capillary column heater (ASI, Pompton Plains, NJ). Solvent B was 0.1% FA, 5% DMSO in water, and Solvent C 0.1% FA, 5% DMSO in ACN. The Q Exactive HF-X was used to analyze the phosphopeptides and TMT labeled peptides. For the phosphopeptides, the gradient was 4–15% C in 40 min, followed by 15–28% C in 20 min. For the analysis of TMT labeled peptides, the gradient was 8–34% C in 15 min. The MS1 and MS2 parameters were kept the same as for the micro-flow LC–MS/MS system. Peptides from kinobeads sample were separated by the same nLC system using a linear gradient from 5 to 33% C in 52 min. MS1 spectra were acquired at a resolution of 60,000 (at m/z 200) in the Orbitrap using a maximum IT of 10 ms and an automatic gain control (AGC) target value of 3E6. For MS2 spectra, up to 12 peptide precursors were isolated for fragmentation (isolation width of 1.7 Th, maximum IT of 75 ms, AGC value of 2e5). Precursors were fragmented by HCD using 25% normalized collision energy (NCE) and analyzed in the Orbitrap at a resolution of 15,000. The dynamic exclusion duration of fragmented precursor ions was set to 30 s.

**Data processing and analysis.** Except otherwise noted, the data were searched by MaxQuant v1.6.2.3[52] against the UniProtKB Human Reference Proteome database (v22.07.13, 88,381 entries). Default MaxQuant parameters were used. Trypsin was specified as the enzyme, cleaving after all lysine and arginine residues and allowing up to two missed cleavages. Carbamidomethylation of cysteine was specified as fixed modification and protein N-terminal acetylation and oxidation of methionine were considered as variable modifications. The false discovery rate (FDR) was set to 1% on the site, peptide-spectrum match (PSM) and protein levels. For the 1,550 injection performance test, the datasets of HeLa (200 raw files), urine (200 raw files), CSF (200 raw files), plasma (200 raw files), and PROCAL samples (270 raw files) were separately searched against the UniProtKB Human Reference Proteome database (v22.07.13, 88,381 entries) supplemented with PROCAL peptide sequences[32]. The 480 raw files of the placenta sample were searched against the same UniProtKB Human Reference Proteome database without PROCAL peptide sequences. The match-between-runs feature was enabled, the matching and alignment time window was set to 0.7 min and 5 min. Label-free quantification was enabled and LFQ min ratio count was set to 2. For re-analysis of published datasets, we downloaded the original raw data files and re-searched them by MaxQuant as above.

For the TMT data, batch specific TMT correction factors (product number: 90406 and A34807) were added as specific parameters. The TMT data generated both by MS2 and MS3 methods were searched together but with different parameter groups. The reporter ion mass accuracy was set to the default value of 0.003 Da, and the MS2 tolerance was set to 0.4 Da for IT MS. All other parameters were default. For phosphopeptides, phopho (STY) was set to variable modification.

For Kinobeads samples, peptide and protein quantification was performed using MaxQuant (v. 1.6.0.1) by searching MS2 spectra against all canonical protein sequences as annotated in the Uniprot reference database (human proteins only, 20,230 entries, downloaded 06.07.2017) using the embedded search engine Andromeda. Phopho (STY), oxidation of methionine and N-terminal protein acetylation were set as variable modifications and carbamidomethylated cysteine as fixed modification. All data were filtered for 1% PSM and 1% protein FDR. Label free quantification and match between runs were enabled within MaxQuant.

For the AP and BioID pull-down samples, there were six replicates (2 biological replicates × 3 technical replicates) for each sample. All the raw files were processed with a previously reported pipeline[53]. Briefly, the search results from Mascot and Comet searches were filtered with iProphet ≥ 0.95 and unique peptides ≥ 2. The high confident interactors were defined as proteins with BFDR ≤ 0.01 using the default options in the SAINTexpress software. For the SAINT analysis of the AP-MS dataset, both the 3xFLAG-empty and 3xFLAG-GFP samples were used as controls, and both control and bait datasets were compressed to three experiments. For the SAINT analysis of the BioID-MS dataset, 3xFLAG-empty, empty-BirA*-FLAG and BirA*-FLAG-GFP samples were used as controls. In each case, the controls (12 replicates from two types of control samples for FLAG, and 18 replicates from three types of control samples for BioID, respectively) were compressed to six virtual controls while bait datasets (six replicates per bait) were compressed to three experiments. For the comparison with previously reported high confidence interactions, we obtained the dataset of AP pull-down samples (EIF4A2 and MEPCE) from the supplemental information of published paper[40], and the dataset of the BioID pull-down samples (LMNA, NIFK, and CTNNA1) was downloaded from the Human Cell Map website (https://humancellmap.org/)[42]. For the online STRING analysis of the five baits[43], the line thickness between each protein indicates the strength of the data support, all the active interaction sources including textmining, experiment, databases, co-expression, gene fusion, neighborhood and co-occurrence were used. The minimum required interaction score was set to 0.70 and the maximum number of interactors to show were set no >10 interactors for the 1st shell, and none interactors for the 2nd shell. The structure previews inside the network bubbles were enabled. The fold change (FC)

values of the interactors (over control pulldowns) assessed by SAINT analysis were annotated in the STRING network (Fig. 4b, c and Supplementary Fig. 14, proteins without FC annotation were not identified as high confident interactors in SAINT analysis).

Data analysis downstream of MaxQuant output results was performed in R[54]. The evidence result file was used for the carry-over analysis. After removing all the reverse peptides, the peptide intensities of each raw file were summarized. As albumin is one of the most abundant proteins in the body fluid samples, we kept the peptides matched to potential contaminating proteins for carry-over analysis. The evidence result files were used for the PROCAL retention time (RT) analysis, as we searched the raw files with both human and PROCAL sequences, we firstly removed all the peptides identified from the human database. For technical reasons, three raw files HeLa_P035214_BA1_S00_A00_R11, Plasma_P035250_BE1_S00_A00_R2 and CSF_P035234_BC1_S00_A00_R15, had to be excluded from the retention time analysis. As the MBR search was enabled, there was sometimes more than one retention time value for one peptide in the same raw file. In such cases, only the retention time with the highest intensity value was kept. In addition, peptides whose intensity values were below 5E7 were excluded from the analysis. Regarding the dynamic range analysis of the TMT sample: dynamic range was defined as the ratio of the maximum and minimum intensity values of a peptide in the TMT11 channels representing different human cancer cell lines. If there were zero value channels, the intensity of the channel with smallest value above zero was used as the minimum intensity value. To ensure a fair comparison, only peptides identified in all the four LC–MS/MS experiments were considered. If peptides were represented by more than one MS/MS spectrum within one experiment, we summed their intensities in the respective TMT channel (s) to calculate the ratio. The t-SNE analysis was performed using the Rtsne package[35] and using the label free quantification (LFQ) intensity values without further normalization as input.

**Reporting summary**. Further information on research design is available in the Nature Research Reporting Summary linked to this article.

## Data availability

The mass spectrometry proteomics data and MaxQuant search results have been deposited with the ProteomeXchange Consortium via the PRIDE[55] partner repository with the dataset identifier PXD015087. The source data underlying Figs. 1b, f, g, 2b, 3b, and 5d–e, and Supplementary Figs. 1a, b, 2c, d, 3c, d, 5a-c, e-i, 6a-c, e-i, 7c, 11c and 13b are provided as Source Data file. The detailed relationship of the original MaxQuant search results information to each plot is also listed in the Supplementary Data 4. All other data are available from the corresponding author on request.

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

## Acknowledgements
Y.B. is grateful for postdoctoral fellowships by the Alexander von Humboldt Foundation, the Carl Friedrich von Siemens Foundation and the National Natural Science Foundation of China (81600046). R.Z. is grateful for a PhD scholarship by the Chinese Scholarship Council (CSC). Parts of this work were funded by grants from the German Science foundation (DFG-SFB1309), the ProteomeTools project (BMBF; grant no. 031L0008A) and the German Excellence Initiative. Work in ACG's laboratory is supported by a Canadian Institutes of Health Research Foundation Grant (CIHR FDN 143301) and the Government of Ontario and Genome Canada and Ontario Genomics (OGI-139). The authors wish to thank Christopher D. Go for sharing the BioID cell lines. We wish to thank all members of the Kuster group for technical assistance and fruitful discussions.

## Author contributions
B.K. conceived the study. O.B., Y.B., and R.Z. set up and optimized the micro-flow LC system. Y.B., R.Z., F.P.B., A.-C.G., and B.K. designed experiments. Y.B., R.Z., F.P.B., C.W., Y.-C.C., M.R., S.W., and D.P.Z. performed experiments. Y.B., C.M., C.W., F.P.B., M.R., J.Z., S.H., J.S., B.H., M.B., and A.-C.G. analyzed data. Y.B. and B.K. wrote the manuscript.

## Competing interests
B.K. is a founder and shareholder of OmicScouts and msAId. D.P.Z. is a founder and shareholder of msAId. They have no operational role in either company. O.B. and M.B. are employees of Thermo Fisher Scientific, R.Z. is currently an employee of Thermo Fisher Scientific. The other authors declare no competing interests.
