## [Peer Review File · Nature Communications]

Reviewers' comments:

Reviewer #1 (Remarks to the Author):

The study by Bian et al. aims to evaluate the utility of so-called high-flow or normal-flow chromatography for bottom-up proteomic analyses. The authors describe the development of the online LC-MS/MS platform using a 1 mm ID stainless steel column and the flow rate of 50 μ L/min. To assess robustness and reproducibility of the method, they also designed a long-term experiment comprised of multiple cycles of single-shot and in-depth analyses (>2,000 samples in total). They reported that the high-flow system can overcome limitations of nLC by improving robustness, throughput, and reproducibility with only a moderate loss of sensitivity and the number of peptide and protein identifications. Additionally, the high-flow setup was tested for analyses of drug-protein interactions, protein-protein interactions, and phosphopeptides, showing its usefulness for a wide range of applications.

The study is technically strong, and its conclusions are convincing. Unfortunately, there have been a number of earlier publications reporting extremely similar observations. For example, the study by Lenčo et al. entitled "Conventional-Flow Liquid Chromatography–Mass Spectrometry for Exploratory Bottom-Up Proteomic Analyses" (Analytical Chemistry 2018) reported identification of 2,835 proteins in a 60 min analysis from 2 μ g HeLa digest using an online LC-MS/MS method with a 1 mm ID column and 68 μ L/min flow rate. Similarly to the manuscript under review, Lenčo et al. also reported the addition of 3% DMSO positively impacts spray stability and the number of peptide identifications. Likewise, Fernández-Niño et al (Front Bioeng Biotechnol. 2015) and Yin et al. (Circ Cardiovasc Genet. 2017) reported that using standard-flow or high-flow LC systems increases throughput, robustness, and reproducibility of bottom-up proteomic analyses – the very same observations that the Bian et al. report here.

I am only left to assume Bian et al. are unaware of this prior work as these high relevant publications are neither acknowledged nor cited. Note there are many other high-flow LC applications for targeted proteomics (Percy et al Methods 2015; Chambers et al MCP 2015, Percy et al Proteomics 2016).

Although the manuscript under consideration extends the utility of the high-flow LC approach to a wider range of bottom-up proteomics sample types, this approach has already been described. As such the manuscript doesn't appear to fit the publishing standards of Nature Communications but could be publishable in a more technical journal with the points below incorporated.

Comments:

1) There are numerous supplementary figures, and it is not clear how each figure relates to the rest of the manuscript or enhances reader's understanding of the work done. Given the manuscript is written in the Brief Communication Reports format, the authors should consider reducing and simplifying their figures, perhaps by removing ones that are not fully described in the text of the article.

2) To demonstrate that the high-flow setup is capable of doing reproducible quantitation, Figure 2 and Supplementary Figure 8 show coefficient of variations (CVs) of quantified proteins. In general, protein-level quantification tends to exhibit less variation because many peptide measurements are averaged to produce a single protein abundance measurement. It would be more convincing to display peptide CVs, as peptide-level quantification is more directly related to chromatographic performance. In addition, quantification of phosphorylation sites is done on the peptide level and is usually quite challenging (Hogrebe et al Nat Com 2018). Demonstrating that using high-flow chromatography positively impacts the quality of phosphopeptide quantification would be interesting and valuable.

3) Typically 8-10 points across a peak are needed to achieve high quality quantitation. Given relatively long duty cycle of Q Exactives, extremely narrow peaks of 2-3 s generated by the shorter gradients may be composed of only a few points, likely adversely impacting quantification. Showing some actually chromatographic peaks, in addition to box plots of average peak widths, and commenting on our ability to quantify such narrow peaks using current state-of-the-art MS instruments would be useful and interesting.

Reviewer #2 (Remarks to the Author):

The manuscript "High-flow chromatography for reproducible and high-throughput quantification of proteomes" by Y. Bian et al. summarizes a large-scale study on the applicability of micro-flow LC to the analysis of proteomic samples usually analyzed by nano-LC. This study represents very useful reference material for proteomic practitioners, especially those working in clinical proteomic labs, i.e. where relatively large amount of protein samples is readily available. This work will definitely provide a significant impact on the field of proteomics: e.g. it will simplify the use of LC-MS equipment for many users and facilitate wider clinical applications. The manuscript is well written and contains staggering amount of experimental material. Experimental section contains sufficient details to reproduce it in any lab. I recommend accepting this manuscript for publications following minor corrections as suggested below.

Major comment:

1. I believe that many chromatographers will disagree with defining the method described here as “high-flow” LC-MS/MS. While the boundaries for distinguishing between normal/micro/nano – flow applications are indeed loosely defined, I think that 50 μ L/min flow rate with 1 mm ID column is definitely falls into the micro-flow category.

There are many generations of HPLC users with different backgrounds and experience level. Some researchers grew up on normal-flow applications (0.5- 1 - 2 mL/min), with slow transition to micro in the 1990’s promoted by desire to save on the cost of the consumables and the availability of new high precision equipment. Others (such as proteomics researchers) started right away with nano-flow - applications driven by the limitations in sample amount. Some of the later ones, being transferred into the field of translational research and clinical applications, have to use micro-flow methods, which are extremely robust. All instrumental issues with micro-flow applications have been resolved, and commercial LC instrumentation is readily available. These researchers may feel that they are dealing with high-flow applications, while the former group just got to the same point from the other side of the flow rate spectrum and will be confused by such a definition.

There is one thing I definitely agree with the authors, that the transfer from nano to micro would not be possible without the latest advancements in MS technology giving better detection sensitivity. The authors have to be praised for being one of the first not to only realize that, but perform a methodical study to confirm this by analyzing variety of usual proteomics samples using this higher-flow approach.

- I believe that the manuscript will benefit from small modification of the first sentences of introduction to provide historical development point of view from normal –to micro and nano-flow (in proteomics era) and now coming back to micro-flow for robust quantitative applications in proteomics.

- I suggest removing “high-flow” where possible or substitute it with “micro-flow”. E.g. manuscript title could be changed as following: “Robust micro-flow chromatography for reproducible high-throughput quantification of proteomes”. I believe that such a change will not take away any novelty aspect presented here.

Minor corrections:

Page 2, line 3 “..injections of cell line, tissue, body fluid, affinity- and phospho-proteomes demonstrate excellent chromatographic..” should have noted the type of analyzed samples – proteolytic digests: e.g.

“..injections of the digests of cell line, tissue, body fluid, affinity- and phospho-proteomes demonstrate excellent chromatographic..”

Page 2, line 5 “..chromatographic (<0.3 % coefficient of variation, CV)..” – maybe unclear for some readers what LC reproducibility is.

Maybe “..chromatographic (<0.3 % coefficient of variation, CV of retention time)..”

Page 2, line 14 “...limit the quantitative reproducibility...” – “limit the reproducibility of protein quantitation...”

Page 2, line 15 “...high complexity or dynamic range” - “...high complexity or wide dynamic range of protein concentrations”

Page 2 line 23-24. “using standard 1 x 150 mm HPLC” – remove “standard”.

And I suggest to remove “(termed high-flow LC-MS/MS)”.

Page 3 line 1 “quantitative reproducibility” – just “reproducibility

Page 3 line 2 “high-flow approach” – “micro-flow” – I suggest making respective changes throughout the manuscript.

Page 3 line 5. “While the wider bore efficiently focusses the analyte on the head of the column (promoting separation), ...” – the meaning of this statement is unclear: “wider bore” cannot focus analytes, and how this “promotes separation”? correct “focusses”

Maybe: “While the larger column diameter improves separation efficiency by eliminating column overloading effects, the higher...”

Page 3. Line 13. What is “trap-elute setup”? does this mean using a trap column? A reader would need to comb through the experimental section to understand that. I suggest to replace it with usual “..compared to the usual setup of nLC..” – this will cover both popular nano-flow setups (with and without trap column).

Page 3. “...as reported in the recent nLC literature^{10, 11} using only 2-5 times..” – replace with “...as reported recently for nLC-MS/MS setup^{10, 11} using only 2-5 times..”

Page 4 line 3. “Owing to the relative low capacity of the column...” I think this statement is not correct and somewhat contradicts to the previous “wide bore efficiently...”. I believe that lower

carryovers found in this system could be the consequence of overall lower absolute sensitivity of micro-flow ESI ionization and relatively high volume of eluents passing through the system enabling faster mass exchange compared to nano-flow.

I would modify it simply as: "We found extremely low carry-over (average...)"

Page 4. "...led to very high quantitative reproducibility.." – "...led to very high reproducibility of protein quantitation.."

"from 5 ug digest" – "from 5 ug of digest"

"...readily obtained from humans..." - "...readily available for biological fluid samples..."

Page 4 "potential of this set-up for clinical translation" – "potential of this set-up for implementation in clinical laboratories."

Page 5. Sentence starting with "When multiplexing digests from..." should be re-written. E.g. "Tandem mass tags (TMT) quantitation was performed by labeling digests of 11 human cancer cell lines followed by off-line fractionation (48 fractions) and resulted in identification of >7,800 and 6,400 proteins using MS2 (HF-X) and MS3 (Lumos) methods, respectively, from 250 ug of peptides within 16 h of LC-MS time."

Page 5. Line 14: "While such dynamic range is suitable for many applications, larger differences e. g. between cell lines can generally not be reliably detected." The meaning of second part of this sentence is unclear. I think this sentence can be removed without any serious consequences.

Page 5. Line 24. "... because these samples contain a much complexity-reduced proteome¹⁸ " – simply : "...due to lower complexity of these samples.."

Page 6, The sentence starting with "Somewhat surprisingly..." should be re-written. E.g. "2 mg of HeLa digest was fractionated into 12 pulled fractions, followed by IMAC phosphopeptide enrichment and LC-MS analysis. Somewhat surprisingly given the overall low abundance of phosphopeptides (typically ~1% relative to the total), micro-flow LC-MS identified 32,493 unique phosphopeptides within 12 h of gradient time."

Page 6. Line 13. "...only commercially components are required..." – this is not sufficiently clear. The only "commercial component", which is missing in nano-LC set-up is the column(s). Even this isn't

entirely correct: there are sufficiently robust commercially available nano-columns (e.g. EasySpray from Thermo).

I suggest to modify it as following:

“...performance characteristics, and availability of high quality micro-flow columns, the authors expect..”

Experimental.

I suggest some revision of experimental portion to make it as clean as major portion of the manuscript:

Page 12, Experimental. Will having the excess of alkylating CAA agent lead to peptide overalkylation?

Page 15. Line 19. “A sample loop with 20 ul of volume was installed on the injection valve, and the sample loop was kept in the LC connection line during the gradient. The column was put into the integrated column oven.” – simply “20 uL sample loop was used throughout the micro-flow experiments in direct injection mode.”

Column temperature is mentioned on the next page: it is not necessary to use it here.

Page 16. Line 1. There is no need to describe here Halo peptide ES-C18 experiments.

Page 16. Line 6. Remove “...and samples were directly injected onto the head of the column.” - see comment for page 15 above.

Page 16. Line 6. In my opinion, the whole section (3 sentences) starting with “As a side note...” has nothing to do with the manuscript content.

Page 16-17. Description of pumps’ performance and set up has lots of details, however the most important ones are missing: what type of Vanquish pump (I assume binary gradient) and eluent mixer option have been used for the high-throughput experiments? Chromatographic profile in Fig. 1 show very small delay volume of ~125 ul and given approximate 80 uL dead volume of the column, the mixer’s volume was extremely low. What was the volume of the mixer installed on the pump?

These details are critical in assisting the set up of system like this in the other labs.

Couple of questions for Supplementary Figures:

Fig S1 c. No DMSO blue LC profile is not visible behind red DMSO plot. Showing relative abundance scale without giving maximum signals (NL values) makes it impossible to compare no-DMSO/DMSO detection sensitivity.

Figure S2 c. The chromatographic peak width is drastically different for identical gradient conditions with different MS acquisition speed. E.g. panel c for 50-100-200-500 ng injections using 28 and 41 Hz settings. This should not be the case, unless something happened to the LC system between these runs.

Supplemenatry Tables 2, 3, 4 did not show up correctly after conversion into pdf format. At the same time they look ok in Excel.

Sincerely yours,

Oleg Krokhin, PhD

Point to point response to reviewer comments

The authors thank the reviewers for their insightful comments which we took into account when revising the manuscript. Following discussions with the editorial office, we changed the manuscript format from a Brief Communication to a Research Article which allowed us to use substantially more space in order to present the work. Together with the response to the reviewer comments, we believe that the manuscript has become much stronger.

Reviewer #1 (Remarks to the Author):

The study by Bian et al. aims to evaluate the utility of so-called high-flow or normal-flow chromatography for bottom-up proteomic analyses. The authors describe the development of the online LC-MS/MS platform using a 1 mm ID stainless steel column and the flow rate of 50 uL/min. To assess robustness and reproducibility of the method, they also designed a long-term experiment comprised of multiple cycles of single-shot and in-depth analyses (>2,000 samples in total). They reported that the high-flow system can overcome limitations of nLC by improving robustness, throughput, and reproducibility with only a moderate loss of sensitivity and the number of peptide and protein identifications. Additionally, the high-flow setup was tested for analyses of drug-protein interactions, protein-protein interactions, and phosphopeptides, showing its usefulness for a wide range of applications.

The study is technically strong, and its conclusions are convincing. Unfortunately, there have been a number of earlier publications reporting extremely similar observations. For example, the study by Lenčo et al. entitled “Conventional-Flow Liquid Chromatography–Mass Spectrometry for Exploratory Bottom-Up Proteomic Analyses” (Analytical Chemistry 2018) reported identification of 2,835 proteins in a 60 min analysis from 2 ug HeLa digest using an online LC-MS/MS method with a 1 mm ID column and 68 uL/min flow rate.

Similarly to the manuscript under review, Lenčo et al. also reported the addition of 3% DMSO positively impacts spray stability and the number of peptide identifications. Likewise, Fernández-Niño et al (Front Bioeng Biotechnol. 2015) and Yin et al. (Circ Cardiovasc Genet. 2017) reported that using standard-flow or high-flow LC systems increases throughput, robustness, and reproducibility of bottom-up proteomic analyses – the very same observations that the Bian et al. report here. I am only left to assume Bian et al. are unaware of this prior work as these high relevant publications are neither acknowledged nor cited. Note there are many other high-flow LC applications for targeted proteomics (Percy et al Methods 2015; Chambers et al MCP 2015, Percy et al Proteomics 2016).

The authors are happy to read that the reviewer considers the work technically strong and the conclusions convincing.

As far as the prior literature is concerned, we note that the extremely short format of a “Brief Communication” did not allow us to cover all the relevant literature. Now that the manuscript has been formatted into a research article, we were able to expand the introduction to give more credit to that prior work. We would like to point out that the original manuscript already cited the work of Lenčo et al. (as Ref. 9) and we explicitly stated in the text that we based our study on this prior work. As part of the original manuscript, we also stated that we first reproduced the results of Lenčo et al. before making our modifications and demonstrating the merits of the approach by

several applications and a lot of data. In light of the comments made by reviewer #2, we dropped part of the manuscript that ‘merely’ reproduced the results of Lenčo et al. We further note that our laboratory introduced the use of DMSO in LC solvents to boost ESI response for proteomic applications (Hahne et al. Nat Methods 2013). Hence, Lenčo et al. (as well as many others) followed our lead not *vice versa*.

In the revised manuscript, we now cite the work of Fernández-Niño et al. (in the context of discovery proteomics) who noted already in 2015 that standard LC-MS/MS is an option for samples where available quantities are not limiting. In this prior work, ~4,000 peptides from ~800 *E. coli* proteins were identified from 40 µg injected peptides and using 120 min gradient time. The current work improves this more than 100x in terms of sample quantity requirements and 4-8x in terms of analysis time required both of which very substantially extend the utility of ‘higher-flow’ LC configurations for proteomic applications.

We now also refer to the other publications mentioned by the reviewer (Yin et al. Circ Cardiovasc Genet. 2017; Percy et al Methods 2015; Chambers et al MCP 2015, Percy et al Proteomics 2016) in the context of targeted proteomics (MRM/SRM) where larger i. d. columns have long been used to improve the quantitative performance LC-MS/MS, albeit for a small number of pre-defined peptides and taking advantage of the often higher sensitivity of SRM/MRM type experiments. Again, we believe that our contribution goes substantially beyond this prior work as it demonstrates applicability also to a wide range of discovery proteomics applications.

Although the manuscript under consideration extends the utility of the high-flow LC approach to a wider range of bottom-up proteomics sample types, this approach has already been described. As such the manuscript doesn't appear to fit the publishing standards of Nature Communications but could be publishable in a more technical journal with the points below incorporated.

The authors acknowledge that we are not the first to show in principle that higher flow LC separations can be used in discovery proteomics and we now cover more of the prior literatures as mentioned above. However, we think that our study also goes substantially beyond the prior literature in that it actually demonstrates the claims of throughput and robustness by a very large amount of data (thousands of runs) and claims of applicability to discovery proteomics by a wide range of examples (body fluids, tissue, cell lines, affinity purifications, phosphoproteomes). In light of the substantially expanded revised manuscript, we are confident that the impact of our work on the field could be more profound. It has certainly transformed the authors’ laboratories as it is now possible to take on discovery proteomics projects of a scale that would not be possible or even conceivable using nano-LC-MS/MS.

Comments:

1) There are numerous supplementary figures, and it is not clear how each figure relates to the rest of the manuscript or enhances reader's understanding of the work done. Given the manuscript is written in the Brief Communication Reports format, the authors should consider reducing and simplifying their figures, perhaps by removing ones that are not fully described in the text of the article.

The authors agree that the Brief Communication was somewhat overloaded with information. By expanding the manuscript, we think that is now much clearer how supplementary figures relate to the main text and figures. The intent of showing many supplementary figures was and is to provide the supporting evidence of points we discuss in the main manuscript.

2) To demonstrate that the high-flow setup is capable of doing reproducible quantitation, Figure 2 and Supplementary Figure 8 show coefficient of variations (CVs) of quantified proteins. In general, protein-level quantification tends to exhibit less variation because many peptide measurements are averaged to produce a single protein abundance measurement. It would be more convincing to display peptide CVs, as peptide-level quantification is more directly related to chromatographic performance. In addition, quantification of phosphorylation sites is done on the peptide level and is usually quite challenging (Hogrebe et al Nat Com 2018). Demonstrating that using high-flow chromatography positively impacts the quality of phosphopeptide quantification would be interesting and valuable.

As suggested, we have added the peptide level quantification data for the body fluids and HeLa cells to the main figure (new Fig. 3d). As expected, CV values at the peptide level are higher than the ones on the protein level, but 75-85% of all peptides still have CVs below 20% for all types of samples. In addition, the variation within and across cycles at the peptide level was still small enough to clearly separate the plasma samples of the five individuals, showing that the micro-flow system enables reproducible quantification.

Regarding the quantification of phosphopeptides, we have made two additions to the manuscript. First, we show dose response curves for 4 phosphopeptides (new Fig. 4f) of the kinase GSK3A as well as the aggregated data on the protein level in response to the inhibitor AT-9283. The data shows that the EC₅₀ values on the protein and peptide level are very similar, further supporting that quantification at the (phospho)peptide level is possible. Second, we have added examples for phosphorylation-site isomers as Fig. 5f and Supplementary Fig. 16. While the micro-flow data does produce sharper peaks, the nano-flow system also separates the site isomers. A clear advantage of the micro-flow separation is therefore not generally obvious. We have added more text to the revised manuscript and maintain our conservative conclusion that while phosphoproteome analysis is surprisingly well possible, nano-LC still plays out its advantages in terms of sensitivity.

3) Typically 8-10 points across a peak are needed to achieve high quality quantitation. Given relatively long duty cycle of Q Exactives, extremely narrow peaks of 2-3 s generated by the shorter gradients may be composed of only a few points, likely adversely impacting quantification. Showing some actually chromatographic peaks, in addition to box plots of average peak widths, and commenting on our ability to quantify such narrow peaks using current state-of-the-art MS instruments would be useful and interesting.

The authors agree that this point requires careful consideration and it was indeed part of the method development. We decided to use a “28 Hz MS data acquisition method” throughout the manuscript. In this method, MS1 spectra are collected at 60,000 resolution and we allowed a maximum of 12 MS2 spectra at 15,000 resolution per cycle and a maximum precursor injection time of 22 ms. The resulting maximum cycle time between two MS1 spectra was about 0.6 seconds (often shorter due to the fact that not every cycle actually triggered 12 MS2 spectra). A cycle time of 0.6 seconds requires an LC peak width at base of 4.8 to 6 seconds to obtain 8-10 data points for quantification. The table below shows that actual median LC peak widths at base were ~8 seconds for identified peptides (corresponding to at least 13 MS1 data points across the LC peak) even for 15 min gradients compared to ~5 seconds for MS1 features that were detected but not identified (presumably because of low MS intensity). This strongly indicates that most peptides that can be identified, can also be quantified based on a sufficient number of MS1 data points. We chose the drug-protein interaction data for this analysis because of the fact that there are actual quantitative changes in proteins in that experiment.

Experiment	All peaks (s)	Identified peaks (s)	Non-identified peaks (s)
DMSO	5.4	8.0	5.0
3 nM	5.4	8.0	5.0
10 nM	5.5	8.1	5.3
30 nM	5.5	8.0	5.2
100 nM	5.5	8.1	5.3
300 nM	5.4	8.0	5.3
1000 nM	5.4	8.0	5.2
3000 nM	5.4	8.0	5.0
30000 nM	5.4	7.9	5.3
PDPD	5.4	7.9	5.2

Table R1. Distribution of the median retention length (the LC peak width at base in the “allPeptides.txt” file of MaxQuant search results, in seconds) of MS1 features (peaks) detected in 15 min gradients. The data was extracted from the micro-flow LC-MS/MS analysis of the kinobeads pulldown samples. The MS cycle time was less than 0.6 second.

As suggested, we have added several examples of extracted ion chromatograms to the manuscript. First, we show in main Fig. 4e (drug-protein interaction example, 15 min LC gradient) that there are sufficient data points at every drug dose to quantify the change of the peptide QWALEDIFEIGRPLGK corresponding to the kinase AURKA (a target of the drug AT-9283 used in this experiment). Second, we have added extracted ion chromatograms of phosphopeptides to Fig 5f (and Supplementary Fig. 16) again showing by example that the LC peak is generally well represented.

Reviewer #2 (Remarks to the Author):

The manuscript “High-flow chromatography for reproducible and high-throughput quantification of proteomes” by Y. Bian et al. summarizes a large-scale study on the applicability of micro-flow LC to the analysis of proteomic samples usually analyzed by nano-LC. This study represents very useful reference material for proteomic practitioners, especially those working in clinical proteomic labs, i.e. where relatively large amount of protein samples is readily available. This work will definitely provide a significant impact on the field of proteomics: e.g. it will simplify the use of LC-MS equipment for many users and facilitate wider clinical applications. The manuscript is well written and contains staggering amount of experimental material. Experimental section contains sufficient details to reproduce it in any lab. I recommend accepting this manuscript for publications following minor corrections as suggested below.

The authors are happy to read that the reviewer values the amount of work that went into this study and thinks the work will have significant impact on the field

Major comment:

1. I believe that many chromatographers will disagree with defining the method described here as “high-flow” LC-MS/MS. While the boundaries for distinguishing between normal/micro/nano – flow applications are indeed loosely defined, I think that 50 uL/min flow rate with 1 mm ID column is definitely falls into the micro-flow category.

The authors acknowledge that micro-flow is the more widely accepted term and we changed the naming to micro-flow for clarity throughout the manuscript.

There are many generations of HPLC users with different backgrounds and experience level. Some researchers grew up on normal-flow applications (0.5- 1 - 2 mL/min), with slow transition to micro in the 1990's promoted by desire to save on the cost of the consumables and the availability of new high precision equipment. Others (such as proteomics researchers) started right away with nano-flow - applications driven by the limitations in sample amount. Some of the later ones, being transferred into the field of translational research and clinical applications, have to use micro-flow methods, which are extremely robust. All instrumental issues with micro-flow applications have been resolved, and commercial LC instrumentation is readily available. These researchers may feel that they are dealing with high-flow applications, while the former group just got to the same point from the other side of the flow rate spectrum and will be confused by such a definition.

There is one thing I definitely agree with the authors, that the transfer from nano to micro would not be possible without the latest advancements in MS technology giving better detection sensitivity. The authors have to be praised for being one of the first not to only realize that, but perform a methodical study to confirm this by analyzing variety of usual proteomics samples using this higher-flow approach.

The authors are grateful that one of the main points of the paper has come across that clearly. We indeed think that now is the time to move many proteomic applications to micro-flow in order to increase the ease, quality and throughput of proteomic experiments.

- I believe that the manuscript will benefit from small modification of the first sentences of introduction to provide historical development point of view from normal –to micro and nano-flow (in proteomics era) and now coming back to micro-flow for robust quantitative applications in proteomics.

Please also see our response to reviewer #1. We have expanded the introduction to reflect that micro-flow LC-MS/MS is already successfully and routinely used in targeted proteomics and that it is increasingly used for DIA. We now also provide more coverage of the prior literatures regarding the use of micro-flow LC-MS/MS in discovery proteomics.

- I suggest removing "high-flow" where possible or substitute it with "micro-flow". E.g. manuscript title could be changed as following: "Robust micro-flow chromatography for reproducible high-throughput quantification of proteomes". I believe that such a change will not take away any novelty aspect presented here.

We have changed high-flow to micro-flow throughout the manuscript and have modified the title accordingly.

Minor corrections:

Page 2, line 3 "...injections of cell line, tissue, body fluid, affinity- and phospho-proteomes demonstrate excellent chromatographic.." should have noted the type of analyzed samples – proteolytic digests: e.g.

"...injections of the digests of cell line, tissue, body fluid, affinity- and phospho-proteomes demonstrate excellent chromatographic.."

We have changed this in the revised manuscript

Page 2, line 5 “..chromatographic (<0.3 % coefficient of variation, CV)..” – maybe unclear for some readers what LC reproducibility is.

Maybe “..chromatographic (<0.3 % coefficient of variation, CV of retention time)..”

We have changed this in the revised manuscript

Page 2, line 14 “...limit the quantitative reproducibility...” – “limit the reproducibility of protein quantitation...”

We have changed this in the revised manuscript

Page 2, line 15 “...high complexity or dynamic range” - “...high complexity or wide dynamic range of protein concentrations”

We have changed this in the revised manuscript

Page 2 line 23-24. “using standard 1 x 150 mm HPLC” – remove “standard”.
And I suggest to remove “(termed high-flow LC-MS/MS)”.

We have changed this in the revised manuscript

Page 3 line 1 “quantitative reproducibility” – just “reproducibility”

We have changed this in the revised manuscript

Page 3 line 2 “high-flow approach” – “micro-flow” – I suggest making respective changes throughout the manuscript.

We have changed this throughout in the revised manuscript

Page 3 line 5. “While the wider bore efficiently focusses the analyte on the head of the column (promoting separation), ...” – the meaning of this statement is unclear: “wider bore” cannot focus analytes, and how this “promotes separation”? correct “focusses”

Maybe: “While the larger column diameter improves separation efficiency by eliminating column overloading effects, the higher...”

We have changed this in the revised manuscript

Page 3. Line 13. What is “trap-elute setup”? does this mean using a trap column? A reader would need to comb through the experimental section to understand that. I suggest to replace it with usual “..compared to the usual setup of nLC..” – this will cover both popular nano-flow setups (with and without trap column).

We have changed this in the revised manuscript as suggested

Page 3. “...as reported in the recent nLC literature^{10, 11} using only 2-5 times..” – replace with “...as reported recently for nLC-MS/MS setup^{10, 11} using only 2-5 times..”

We have rephrased this in the revised manuscript

Page 4 line 3. “Owing to the relative low capacity of the column...” I think this statement is not correct and somewhat contradicts to the previous “wide bore efficiently...”. I believe that lower carryovers found in this system could be the consequence of overall lower absolute sensitivity of micro-flow ESI ionization and relatively high volume of eluents passing through the system enabling faster mass exchange compared to nano-flow.

I would modify it simply as: “We found extremely low carry-over (average...)”

We have rephrased the section but also would like to give readers a reason for observing such low carry-over. “The low carry-over is likely owing to the very low amount of sample loaded on the column relative to its capacity, the high volume of solvents passing over the column and, to some extent, the lower absolute sensitivity of the micro-LC system.”

Page 4. “..led to very high quantitative reproducibility..” – “..led to very high reproducibility of protein quantitation..”

“from 5 ug digest” – “from 5 ug of digest”

“...readily obtained from humans...” - “...readily available for biological fluid samples...”

We have modified these sentences.

Page 4 “potential of this set-up for clinical translation” – “potential of this set-up for implementation in clinical laboratories.”

We have modified this statement.

Page 5. Sentence starting with “When multiplexing digests from...” should be re-written. E.g “Tandem mass tags (TMT) quantitation was performed by labeling digests of 11 human cancer cell lines followed by off-line fractionation (48 fractions) and resulted in identification of >7,800 and 6,400 proteins using MS2 (HF-X) and MS3 (Lumos) methods, respectively, from 250 ug of peptides within 16 h of LC-MS time.”

We have expanded this aspect to an entire manuscript section to improve clarity.

Page 5. Line 14: “While such dynamic range is suitable for many applications, larger differences e. g. between cell lines can generally not be reliably detected.” The meaning of second part of this sentence is unclear. I think this sentence can be removed without any serious consequences.

Please also see our previous comment. We have changed the text to bring our more clearly, that improved chromatography also improves the ability to detect larger expression differences of proteins between cell lines.

Page 5. Line 24. “... because these samples contain a much complexity-reduced proteome¹⁸ “ – simply : “...due to lower complexity of these samples..”

We have rephrased this in the revised manuscript

Page 6, The sentence starting with “Somewhat surprisingly...” should be re-written. E.g. “2 mg of HeLa digest was fractionated into 12 pulled fractions, followed by IMAC phosphopeptide enrichment and LC-MS analysis. Somewhat surprisingly given the overall low abundance of phosphopeptides (typically ~1% relative to the total), micro-flow LC-MS identified 32,493 unique phosphopeptides within 12 h of gradient time.”

We have rephrased this in the revised manuscript largely as suggested.

Page 6. Line 13. “..only commercially components are required...” – this is not sufficiently clear. The only “commercial component”, which is missing in nano-LC set-up is the column(s). Even this isn’t entirely correct: there are sufficiently robust commercially available nano-columns (e.g. EasySpray from Thermo).

I suggest to modify it as following:

“...performance characteristics, and availability of high quality micro-flow columns, the authors expect..”

We have modified these sentences accordingly.

Experimental.

I suggest some revision of experimental portion to make it as clean as major portion of the manuscript:

Page 12, Experimental. Will having the excess of alkylating CAA agent lead to peptide overalkylation?

The short answer is that overalkylation by CAA is minimal. A small excess of CAA over the reducing agent DTT is necessary to quench excess DTT. For many years, proteomics practitioners have used iodoacetamide (IAA) which is a much more reactive alkylating reagent and which can lead to overalkylation. Work by several laboratories have shown that overalkylation can be reduced very effectively by using the much less reactive chloroacetamide (Nielsen et al Nat Methods 2008) which is why we have been using this for many years.

To address the question directly, we have performed an additional MaxQuant search allowing alkylation at many amino acids and the rate of overalkylation was very low (see table below). In fact so low, that the frequency of overalkylation cannot be distinguished from the false discovery rate criterion used in the analysis. We therefore do not consider overalkylation to be an issue.

Table R1. Data summary of the number of overalkylated peptides and all modified peptides in the gradient test experiment, 2 µg HeLa protein digest was injected for each gradient. The alkylation of the N-term, and alkylation of D, E, H, K, S, T, Y were set as variable modifications in the MaxQuant software. Less than 0.7% of all unique peptides in each experiment were potentially overalkylated.

Gradient (min)	Overalkylated sequence	Modified sequence	Overalkylation
10	56	8,238	0.68%
15	81	11,814	0.69%
30	103	20,395	0.51%
60	84	26,249	0.32%
90	78	22,084	0.35%
120	69	17,114	0.40%

Page 15. Line 19. "A sample loop with 20 ul of volume was installed on the injection valve, and the sample loop was kept in the LC connection line during the gradient. The column was put into the integrated column oven." – simply "20 uL sample loop was used throughout the micro-flow experiments in direct injection mode."

Column temperature is mentioned on the next page: it is not necessary to use it here.

We have modified these sentences accordingly.

Page 16. Line 1. There is no need to describe here Halo peptide ES-C18 experiments.

The reason why we mention these experiments is because we wanted to first reproduce the work by others (Lenco et al Anal Chem 2018) before making our modifications to the method and showing our applications. As suggested, we have removed these experiments from the manuscript.

Page 16. Line 6. Remove "...and samples were directly injected onto the head of the column." - see comment for page 15 above.

We have removed this text.

Page 16. Line 6. In my opinion, the whole section (3 sentences) starting with “As a side note...” has nothing to do with the manuscript content.

We included this small section in order to provide guidance to scientists wishing to adopt our method as we found that the chromatographic performance of different column batches were not the same (see figures below). We agree that this is a relatively minor point but also think that people should be aware of batch to batch column variability.

Figure R1: Batch differences for PepMap C18 columns. Left panel: chromatographic peak width (FWHM) distributions of 5 batches of PepMap C18 columns. Right panel: number of unique peptides identified for each batch of columns (5 μ g protein digestion injection; 60 min gradient time. Note that the right box and bar represent a PepMap C18 column packed with 3 μ m material.

Page 16-17. Description of pumps' performance and set up has lots of details, however the most important ones are missing: what type of Vanquish pump (I assume binary gradient) and eluent mixer option have been used for the high-throughput experiments? Chromatographic profile in Fig. 1 show very small delay volume of ~ 125 μ l and given approximate 80 μ l dead volume of the column, the mixer's volume was extremely low. What was the volume of the mixer installed on the pump? These details are critical in assisting the set up of system like this in the other labs.

The authors apologize for the lack of detail. We have now added this information to the manuscript. Briefly: yes, the modified Vanquish pump is a binary gradient pump. The modified Vanquish pump used in the micro-flow LC experiments had technical characteristics similar to standard high-pressure binary gradient pump in NCS-3500RS module (<https://assets.thermofisher.com/TFS-Assets/CMD/Specification-Sheets/PS-71899-LC-UltiMate-3000-RSLCnano-PS71899-EN.pdf>) with a pump delay volume < 25 nL and a maximum pressure 800 bar. The pump had an extended active micro-flow control up to 100 μ L/min. There was no additional mixer installed between the pump outlet and fluidics. All observed delays in elution are associated only with the volume of the column, the injection loop and the capillary connections between pump, column, auto-sampler, injection loop, and HESI probe.

Couple of questions for Supplementary Figures:

Fig S1 c. No DMSO blue LC profile is not visible behind red DMSO plot. Showing relative abundance scale without giving maximum signals (NL values) makes it impossible to compare no-DMSO/DMSO detection sensitivity.

We have modified the colors and displayed the maximum signals (NL values) in the same plot (Supplementary Fig. 1). The most abundant peak in both experiment is a singly charge compound with $m/z = 421.06$ which is why the max. NL value of the chromatogram is nearly the same. For all the other peaks, the 3% DMSO chromatogram showed clearly higher intensity than the 0% DMSO chromatogram.

Figure S2 c. The chromatographic peak width is drastically different for identical gradient conditions with different MS acquisition speed. E.g. panel c for 50-100-200-500 ng injections using 28 and 41 Hz settings. This should not be the case, unless something happened to the LC system between these runs.

To clarify, the chromatographic peak width distributions only become large at low sample loadings. The apparent discrepancy you refer arises from two issues: a) at very low sample loadings, very few peptides are detected (see table below) which renders the boxes and medians statistically unreliable; b) related, the 28Hz method is much more sensitive in MS/MS mode than the 41 Hz method and thus identifies more peptides and, therefore, the peak widths remain more stable at lower sample quantities. The MS1 acquisition parameters and the cycle time between two MS1 scans were identical between the 28 Hz and the 41 Hz methods.

Table R2: Data summary of the number of identified peaks in the dilution test experiment with different gradients and MS scan speeds.

Injection amount	No. of identified peaks			
	30min_28Hz	30min_41Hz	60min_28Hz	60min_41Hz
0 ng	1	1	1	1
1 ng	1	1	1	1
2 ng	1	1	1	1
5 ng	1	2	1	1
10 ng	161	3	44	6
20 ng	356	3	215	13
50 ng	696	76	364	33
100 ng	2,202	300	825	55
200 ng	5,854	562	1,701	209
500 ng	14,600	3,048	5,788	632
1 μ g	19,104	7,816	13,831	1,960
2 μ g	21,128	16,203	24,037	5,070
5 μ g	21,820	23,218	32,835	17,925
10 μ g	22,065	25,460	34,884	27,813

Supplemenatry Tables 2, 3, 4 did not show up correctly after conversion into pdf format. At the same time they look ok in Excel.

We submitted Supplemenatry Tables 2, 3, 4 (now, Supplemenatry Tables 1, 2, 3) as excel tables. The conversion to pdf happens automatically during manuscript submission. We have mentioned this issue to the publisher. Most people would anyway want to use the Excel tables, which are also available.

Sincerely yours,

Oleg Krokhin, PhD
Assistant Professor
Department of Internal Medicine, University of Manitoba
Senior Scientist, Manitoba Centre for Proteomics and Systems Biology
799 John Buhler Research Centre, 715 McDermot Avenue
Winnipeg, MB, R3E 3P4
Oleg.Krokhine@umanitoba.ca

REVIEWERS' COMMENTS:

Reviewer #1 (Remarks to the Author):

I have read the response to review. Overall I feel the revision does a much better job placing the prior work in context and in that regard the manuscript is much improved. The bottom line for me, however, is that of novelty. DMSO and this higher flow method have been described already and the question is whether running more samples through that same platform has sufficient novelty for a broad journal such as yours.

Since you have invited the revision you have decided that it does. So given that, I have no other major issues and recommend you move forward.